# PI Kinase-EhGEF2-EhRho5 axis contributes to LPA stimulated macropinocytosis in *Entamoeba histolytica*

**Achala Apte**[1], **Maria Manich**[2], **Elisabeth Labruyère**[2], **Sunando Datta**[1]*

**1** Indian Institute of Science Education and Research, Bhopal, Madhya Pradesh, India, **2** Bioimage Analysis Unit, Institut Pasteur, Paris, France

* sunando@iiserb.ac.in

## Abstract

*Entamoeba histolytica* is a protozoan responsible for several pathologies in humans. Trophozoites breach the intestinal site to enter the bloodstream and thus traverse to a secondary site. Macropinocytosis and phagocytosis, collectively accounting for heterophagy, are the two major processes responsible for sustenance of *Entamoeba histolytica* within the host. Both of these processes require significant rearrangements in the structure to entrap the target. Rho GTPases play an indispensable role in mustering proteins that regulate cytoskeletal remodelling. Unlike phagocytosis which has been studied in extensive detail, information on machinery of macropinocytosis in *E. histolytica* is still limited. In the current study, using site directed mutagenesis and RNAi based silencing, coupled with functional studies, we have demonstrated the involvement of EhRho5 in constitutive and LPA stimulated macropinocytosis. We also report that LPA, a bioactive phospholipid present in the bloodstream of the host, activates EhRho5 and translocates it from cytosol to plasma membrane and endomembrane compartments. Using biochemical and FRAP studies, we established that a PI Kinase acts upstream of EhRho5 in LPA mediated signalling. We further identified EhGEF2 as a guanine nucleotide exchange factor of EhRho5. In the amoebic trophozoites, EhGEF2 depletion leads to reduced macropinocytic efficiency of trophozoites, thus phenocopying its substrate. Upon LPA stimulation, EhGEF2 is found to sequester near the plasma membrane in a wortmannin sensitive fashion, explaining a possible mode for activation of EhRho5 in the amoebic trophozoites. Collectively, we propose that LPA stimulated macropinocytosis in *E. histolytica* is driven by the PI Kinase-EhGEF2-EhRho5 axis.

**Data Availability Statement:** All relevant data are within the manuscript and its Supporting Information files.

**Funding:** The authors received no specific funding for this work.

## Author summary

*Entamoeba histolytica* is an enteric parasite in humans, which leads to various pathologies like dysentery, diarrhoea and abscess formation. Host cells are known to secrete chemokines and growth factors, which are utilized by trophozoites for sustenance and pathogenesis. The sustenance of this parasite within the host requires nutrient uptake, which involves macropinocytosis and phagocytosis. However, the regulation of

**Competing interests:** The authors have declared that no competing interests exist.

macropinocytosis is less explored in *E. histolytica*. We have established for the first time that constitutive as well as LPA stimulated macropinocytosis in amoebic trophozoites functions via PI Kinase-EhGEF2-EhRho5 axis. We also excavated the dynamicity and the spatio-temporal regulation of EhRho5 activity and the associated dynamics in the LPA stimulated cells.

## Introduction

*Entamoeba histolytica* is an intestinal protozoan parasite known for various pathologies in humans, including amoebic dysentery, amoebic colitis and liver abscesses [1]. For sustenance and pathogenesis inside the host, trophozoites utilize heterophagy, which includes macropinocytosis and phagocytosis [2,3]. Both the processes require cytoskeletal rearrangements to entrap the target. Despite major differences in stimulus and size, formation of both macropinosomes and phagosomes share the same regulatory proteins. While phagocytosis in the amoebic trophozoites has been investigated in detail [4–7], studies shedding light on the mechanism of amoebic macropinocytosis are very much limited. This prompted us to investigate the signalling responsible for macropinocytosis. *E. histolytica* is known to take up fluid corresponding to ~15% of its cell volume in 2hrs, showing its high 'drinking' capacity [8]. Macropinocytosis is an evolutionarily conserved mode of bulk endocytosis through which cells uptake extracellular fluid into large, irregularly shaped vesicles called macropinosomes. The uptake starts with ruffling in the plasma membrane, which extends around the fluid [9]. Post internalisation, the macropinocytic vesicles are acidified followed by their maturation [9–12].

Formation of macropinosomes involves major rearrangement of membrane proteins, lipids and actin cytoskeleton to support large deformation of the membrane in a spatially and temporally regulated manner. The protrusions and ruffling formed prior to macropinocytosis are responsible for generation of the force that pushes membrane forward [13,14]. The rearrangement of the cytoskeleton, to facilitate the protrusions, is governed by a cohort of proteins, along with Rho GTPases working as the master regulator. These small GTPases exist in either GTP bound active or GDP bound inactive form, majorly regulated by three proteins—Guanine nucleotide exchange factors (GEF), GTPase activating proteins (GAP) and Guanine nucleotide dissociation inhibitor (GDI). These proteins, together shape the activity zones in cells, where abundance of active Rho population mediates the downstream signalling [15–19].

Typically, macropinocytosis is a clathrin independent constitutive process, but it can also be stimulated by growth factors and extracellular stimuli [20,21]. It has been shown that *Dictyostelium*, a soil dwelling close relative of *E. histolytica*, shows enhanced macropinocytosis in the presence of arginine, lysine, glutamate and metabolisable sugar [22]. Earlier studies have shed light on macropinocytosis [16,23,24] but little is known about the involvement of Rho members during induced macropinocytosis [25]. Requirement of RhoG has been shown for formation of membrane ruffles during growth factor induced macropinocytosis in fibroblasts [26]. Although various studies have reported induction of macropinocytosis by growth factors via activation of small GTPases [16,23,24], studies demonstrating their role in *E. histolytica* are still limited [25]. In *E. histolytica*, ~19 Rho proteins have been identified *in silico*, but only few have been so far attributed for their contribution in amoebic pathogenesis. Constitutively active EhRacA shows defect in cytokinesis and erythrocyte phagocytosis but not macropinocytosis [25]. In a proteomic study, EhRacA, EhRacG, EhRacC and EhRacD have been shown to be associated with phagosomes [27]. While functional roles for EhRacA, EhRacG, EhRacC

have been studied, EhRacD (EHI_012240; EhRho5: Arbitrary nomenclature followed throughout the article; Ref. S3 Table) still remains to be characterised.

While transiting from intestinal site of infection to the extraintestinal secondary infection sites, such as in case of hepatic amoebiasis, amoebic trophozoites enters the bloodstream, where they experience various factors from the host environment and may utilize them for nutrient acquisition and pathogenesis. LPA and PDGF are among such few host factors, produced by platelets which are present abundantly in the bloodstream [28,29] It is well established in cells of mammalian origin that LPA and PDGF selectively activate Rho and Rac subfamily respectively, to elicit downstream signalling [30,31]. Earlier studies have shown the translocation of EhRho1 to vesicular membrane and to some extent to the plasma membrane post treatment with LPA [32]. Also, activated (GTP bound) EhRho1 was detected in the amoebic lysates stimulated with LPA [33]. Similarly, fibronectin has been implicated in regulating Rho family of GTPases and cytoskeleton in *Entamoeba histolytica* [34]. Various examples of induced signalling by Tumor Necrosis Factor, fibronectin, serum have been demonstrated in *E. histolytica* [33,35,36]. LPA stimulated EhRho1 has been reported to regulate Phosphatidylinositol Triphosphate levels via PI 3-Kinase and thus invasive behaviour of trophozoites, whereas TNF induced signalling led to PI3K dependent chemotaxis [33,37,38]. These discrete pieces of information from prior studies, prompted us to decipher the signalling cascade involved during macropinocytosis in *E. histolytica*.

In the current study, we have focused on macropinocytosis, a less studied phenomenon in the pathogen and identified that EhRho5, regulates macropinocytosis in *E. histolytica*. Our results have shown that EhRho5 is activated upon growth factor stimulation via EhGEF2 and enhances the macropinocytic efficiency of trophozoites. We have further demonstrated the mechanism of spatio-temporal regulation of EhRho5 upon growth factor stimulation.

## Results

### EhRho5 translocates to plasma membrane upon LPA stimulation

To study the involvement of EhRho5 in growth factor induced signalling, LPA (Lysophosphatidic acid) and PDGF (platelet derived growth factor) were utilised. These growth factors, majorly produced by platelets, are abundant constituents in the serum [28,31]. Both, LPA and PDGF, induce robust cytoskeletal changes, and act as stimulus for various cellular processes [28,29]. To study if these growth factors are involved in altering the localisation of EhRho5, we generated transgenic trophozoites by overexpressing HA epitope tagged EhRho5 under amoebic cysteine synthase promoter to obtain a near endogenous level of expression [39]. Expression and localisation of HA-EhRho5 was confirmed using Western blotting and immunofluorescence respectively (S1A and S1B Fig). We observed that HA-EhRho5 predominantly localised in the cytosol (S1B Fig). Further, serum starved HA-EhRho5 trophozoites were stimulated with LPA and examined for EhRho5 localisation. We observed that EhRho5 translocated from cytosol to plasma membrane and endomembranes upon LPA stimulation (Figs 1A and 1B and S1C–S1E). We also analysed the HA-EhRho5 expressing population of trophozoites by counting the cells showing membrane associated EhRho5 in the presence and in absence of LPA. We observed that 26.7% cells showed membrane localisation of EhRho5 prior to LPA stimulation, while 73.2% cells exhibited HA-EhRho5 translocation post LPA stimulation (Fig 1C). Our observation was further supported by 3D reconstruction of the z-stacks obtained from images acquired using confocal microscopy (S1F Fig and S1 Video). To verify the translocation to the plasma membrane, LPA stimulated trophozoites were examined for colocalization of HA-EhRho5 and heavy subunit Gal/GalNAc Lectin (Hgl), an established plasma membrane marker in amoebic trophozoites [40]. As evident by the correlation

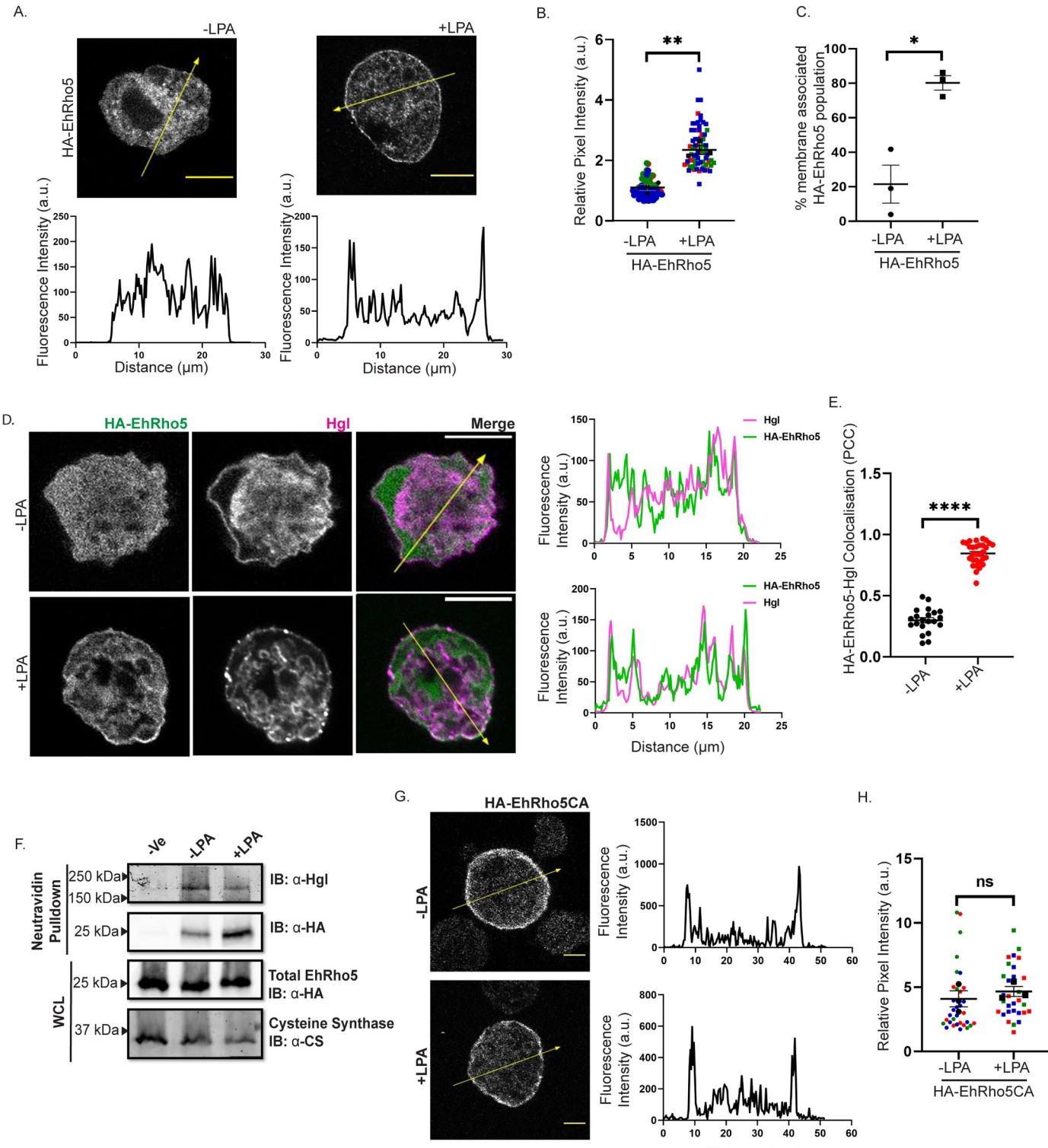

**Fig 1. LPA stimulation leads to EhRho5 translocation.** (A) Serum starved HA-EhRho5 trophozoites were stimulated with 15µM LPA. Cells were fixed and subjected to immunostaining using anti-HA antibody. Images were acquired using a confocal microscope. Line intensity plot shows fluorescence intensity of HA-EhRho5 across the arrow in respective cell (Scale Bar = 10µm). (B) Quantitative analysis of LPA treated cells compared to untreated. A patch of plasma membrane, endomembrane and an adjacent cytosol of the same area were used to measure the pixel intensity. HA-EhRho5 intensity at plasma membrane and endomembrane was normalised with intensity in cytosol to obtain relative pixel intensity for each cell. Values in the SuperPlot are represented as the mean ± SEM of three independent experiments (N = 3, n = 75). Each biological replicate is depicted in one colour. N = Experiments, n = Total no. of cells. Statistical significance

was determined by unpaired Student's *t*-test, **p<0.01. (C) A dot plot shows the percentage of cells exhibiting membrane associated EhRho5 in presence as well as in absence of LPA, in HA-EhRho5 trophozoites. Values are represented as mean ± SEM of three independent experiments (N = 3, n≥80; unpaired Student's *t*-test, *p<0.05). (D) HA-EhRho5 trophozoites were stimulated with or without LPA (15µM), followed by immunostaining with anti-HA and anti-Hgl antibodies (Scale Bar = 10µm). Line intensity plot indicates EhRho5 and Hgl intensities across the arrow. (E) Pearson's correlation coefficient was determined for EhRho5 and Hgl in presence and in absence of LPA and plotted as mean ± SEM (n≥20); unpaired Student's *t*-test, ****p<0.0001. (F) HA-EhRho5 trophozoites stimulated with or without LPA, were subjected to biotinylation to identify membrane bound HA-EhRho5 population. LPA induced cells were labelled with Biotin, and lysed. Cleared lysates were incubated with NeutrAvidin beads (refer Materials and Methods). Bound proteins were resolved using SDS-PAGE followed by immunoblotting with anti-HA, anti-Hgl and anti-CS antibodies. Hgl and cysteine synthase were used as loading controls in the NeutrAvidin bound fraction and whole cell lysate, respectively. (G) HA-EhRho5CA trophozoites were stimulated with or without LPA (15µM), followed by fixation and immunostaining (Scale Bar = 10µm). Line intensity plots indicate the fluorescence intensity of EhRho5CA across the arrow in respective images. (H) SuperPlot shows comparison of relative pixel intensities of HA-EhRho5CA fluorescence at plasma membrane in treated and untreated cells (as mentioned in Fig 1B). Values represent mean ± SEM. N = 3, n≥30; unpaired Student's *t*-test, non-significant (ns), p>0.05.

coefficient (r = 0.84± 0.016), we confirmed that EhRho5 localised on the plasma membrane and endomembranes, post LPA stimulation (Fig 1D and 1E). To confirm the specificity of LPA mediated HA-EhRho5 translocation, we stimulated the trophozoites with PDGF. Unlike LPA, PDGF stimulation did not show any effect on EhRho5 localization (S1G Fig). We further checked the expression status of EhRho5 on transcriptional and translational level upon LPA stimulation (S1H and S1I Fig) and observed no changes (S1H and S1I Fig).

We validated the membrane translocation of EhRho5 using biotinylation assay, a tool extensively used to study the surface population [41,42]. Rho GTPases like the most of the other Ras superfamily of Small GTPases tether to the inner leaflet of plasma membrane by the virtue of an isoprenyl group at the C-terminally located conserved CAAX motif [43]. Therefore, to capture the membrane associated population of EhRho5, we used surface biotinylation approach, where Hgl, an integral membrane protein of *Entamoeba histolytica* served as a positive control for the biotinylation reaction. HA-EhRho5 trophozoites were treated with LPA and EZ-link Sulfo-NHS-biotin. Trophozoites were then lysed and the biotin labelled membrane fraction was captured using NeutrAvidin beads. We observed that post LPA treatment, EhRho5 was enriched in labelled membrane fraction, compared to untreated trophozoites (Figs 1F and S1J).

Rho GTPases are largely known to shuttle between cytosol and plasma membrane, depending on their nucleotide bound state. To examine the nucleotide dependency in LPA mediated EhRho5 translocation, we employed dominant negative (DN) and constitutively active (CA) mutants of EhRho5 (S1K Fig). Though we could establish EhRho5CA expressing cell line, we were unsuccessful in generating EhRho5DN transgenic trophozoites. The instability of the DN form of the GTPases as reported earlier, could be a possible reason for not obtaining the cell line [44]. Serum Starved HA-EhRho5CA trophozoites were stimulated with LPA and localisation of the GTPase mutant was examined. Images obtained by confocal microscopy revealed that EhRho5CA localised on plasma membrane irrespective of LPA treatment (Fig 1G and 1H). Thus, the membrane localization of the GTPase defective, constitutively GTP bound mutant of EhRho5 did not rely on LPA treatment, suggesting a possible correlation of LPA mediated translocation of the GTPase with its nucleotide status.

## LPA stimulated membrane targeting of EhRho5 leads to its activation

Selective translocation of Rho GTPases is known upon stimulation with growth factors like LPA and PDGF [19,45–47]. The membrane translocation has been shown to be coincidental with the activation of the GTPases, which further initiates downstream signalling via binding to effector molecules [46,48,49]. To check whether LPA mediated translocation also leads to activation of EhRho5, we utilised effector pulldown assay. Effectors of Rho GTPases are specific for each of the subfamily, for example, Rhotekin is a known effector of RhoA in

mammalian cells [50]. To identify whether EhRho5 represents a Rho subfamily or Rac subfamily, we investigated its binding to RBD (Rho binding domain of mRhotekin) and PBD (p21 activated kinase binding domain of PAK). GST-RBD and GST-PBD have been frequently used as bait to look for the presence of the activated Rho-GTP population in numerous studies [19,51] including some in *E. histolytica* [52,53]. We cloned EhRho5 in bacterial expression vector pET28a+ with His tag. His-EhRho5, GST-RBD and GST-PBD were expressed in BL21 cells (S2A–S2C Fig). The nucleotide bound to His-EhRho5 was exchanged with excess of GMPPNP, a nonhydrolyzable GTP analogue and GDP as previously published [54]. GST-RBD and GST-PBD immobilised on Sepharose beads, were incubated with His-EhRho5-GMPPNP and His-EhRho5-GDP separately, followed by removal of unbound proteins. Using $\alpha$-His monoclonal antibody, Western blot was carried out to determine the nucleotide dependency of binding. We observed that it is only RBD which responds differentially to the GDP bound inactive EhRho5 from the GMPPNP (GTP analog) bound EhRho5. Based on the definition of a GTPase effector, we therefore concluded that RBD, rather than the PBD, could serve as an *in vitro* effector of EhRho5. Accordingly, we concluded that EhRho5 belongs to the Rho subfamily (Figs 2A and 2B and S2D). To verify the activation status of EhRho5 upon LPA treatment in the trophozoites, a similar strategy was employed [51]. Serum starved HA-EhRho5 trophozoites were stimulated with LPA for 15 mins. GST-RBD and GST-PBD were used to pulldown the active population of EhRho5 from the whole cell lysates of control and LPA stimulated trophozoites. Analysis of Western blot revealed a 7-fold increase in binding of EhRho5 with RBD, post LPA treatment, confirming the activation of the GTPase (Figs 2C and 2D and S2E). Additionally, in HA-EhRho5 expressing trophozoites, the GTPase did not show any binding to GST-PBD further strengthening our conclusion that EhRho5 belongs to the Rho subfamily rather than the Rac subfamily.

Rho GTPases being molecular switches, shuttle between GTP loaded active and GDP loaded inactive form. We have established that within 15 minutes of LPA stimulation, a quantifiable EhRho5-GTP population is observed (Fig 2C and 2D). Therefore, to get more insight into how the dynamics of EhRho5 contribute to activation and translocation of the GTPase observed in steady state, we performed FRAP studies. First, we cloned EhRho5 in a tetracycline inducible plasmid with N-ter GFP tag, allowing us to regulate the expression and follow the dynamics of the GTPase in live trophozoites. GFP-EhRho5 transgenic trophozoites were generated and the localisation was compared with HA-EhRho5 trophozoites (Figs 2E and 2F and S2F–S2I). Although 50% of GFP-EhRho5 trophozoites were already showing localisation of GFP-EhRho5 at plasma membrane and endomembranes, we observed 30% increase in the cell population harbouring membrane associated GFP-EhRho5 post LPA stimulation (Figs 2F and 2G and S2F–S2I), thus further supporting our observation in HA-EhRho5 trophozoites (Figs 1A–1C and S1C–S1E). Then, GFP-EhRho5 trophozoites were stimulated in presence and in absence of LPA, followed by live cell imaging for FRAP. In the analysis, the fluorescence of GFP-EhRho5 was photobleached in patch of plasma membrane and endomembrane. The recovery of fluorescence at the bleached site was monitored by time-lapse imaging on a confocal microscope for 1 min (Fig 2H). The fluorescence recovery could be best approximated using a double exponential model with $\tau_1$ and $\tau_2$ as the characteristic time constant. Out of the two recovery rates, only $\tau_1$ was responsive to LPA stimulation (S2J Fig and S1 Table). Therefore, we had analysed $\tau_1$ in all the conditions of the experiment throughout the study. We observed that GFP-EhRho5 was rapidly exchanged at the plasma membrane with a $\tau_{1/2}$ of 2.8 $\pm$0.2 sec upon LPA stimulation, while unstimulated trophozoites showed a $\tau_{1/2}$ of 3.5$\pm$0.1 sec (S2 Video). At Vesicles, LPA treated and untreated trophozoites showed a $\tau_{1/2}$ of 3.7$\pm$0.2 sec and 4.6$\pm$0.2 sec, respectively (S3 Video). Taken together, we could conclude that LPA

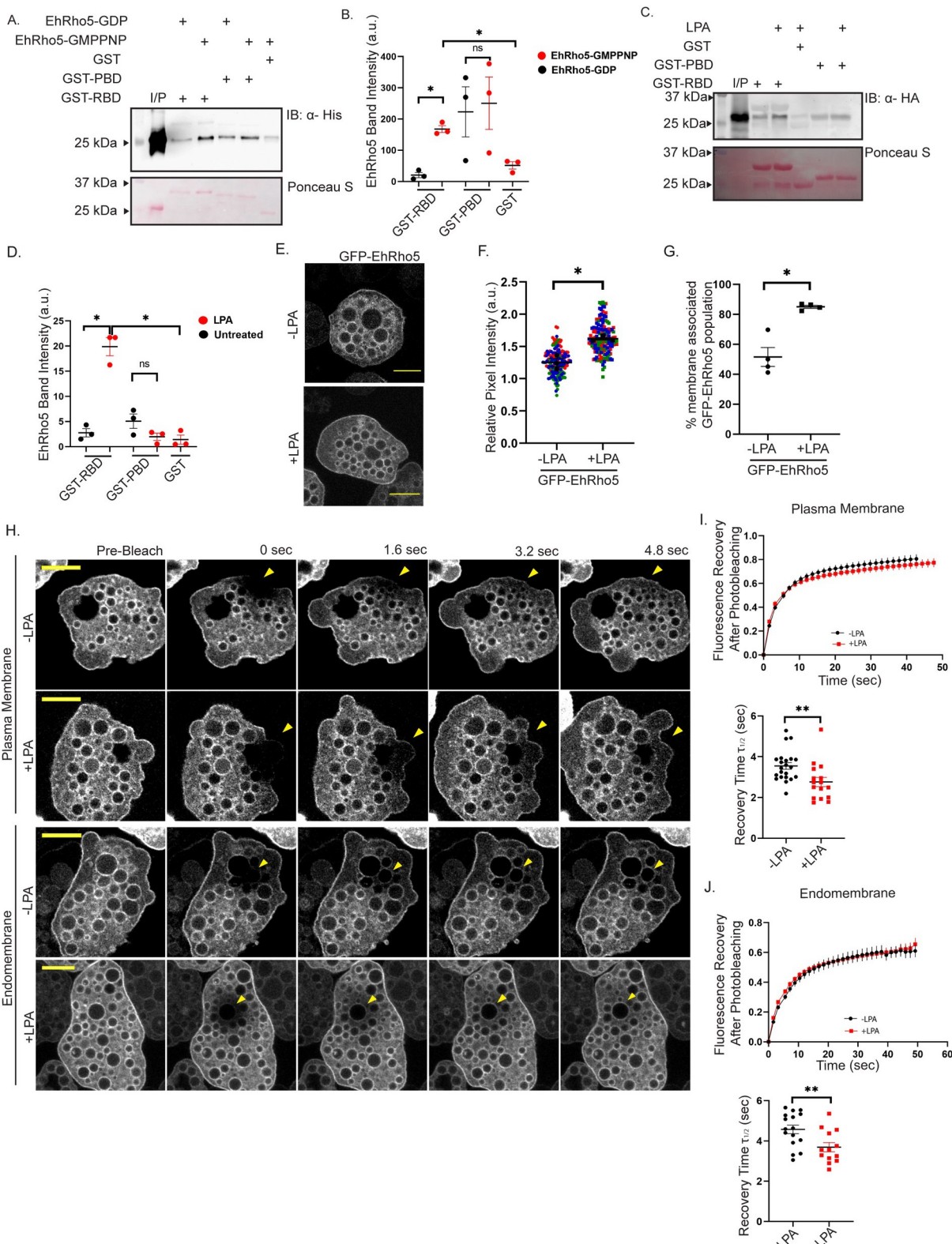

**Fig 2. EhRho5 gets activated upon membrane targeting by LPA.** (A) His-EhRho5 was loaded with GMPPNP and GDP followed by incubation with GST-RBD, GST-PBD and GST bound to glutathione sepharose beads individually for 30mins. Unbound proteins were removed by washing and bound proteins were resolved by SDS-PAGE. Immunoblotting was performed using anti-His antibody. Equal moles

of baits were used in the experiment (ponceau S). (B) Quantitative analysis of the band intensity for His-EhRho5 was performed using ImageJ software. Background intensity was subtracted from His-EhRho5 band intensity and normalisation was done with bait's band intensity to plot mean ± SEM values. N = 3, ratio paired Student's $t$-test, *p<0.05, non-significant (ns), p>0.05. (C) HA-EhRho5 trophozoites were serum starved for 12 hrs and stimulated with LPA. Cells were lysed in lysis buffer containing magnesium chloride. Cleared lysates were incubated with GST-RBD, GST-PBD and GST bound to glutathione-coupled agarose beads for 2hrs. Beads were washed with lysis buffer thrice and bound proteins were resolved by SDS-PAGE. Immunoblotting was performed using anti-HA antibody to catch active EhRho5 population. (D) Active EhRho5 band intensity was determined by background subtractions and normalisation with bait's band intensity. Values are resultant of three independent experiments shown as mean ± SEM. N = 3, ratio paired Student's $t$-test; *p<0.05; non-significant (ns) p>0.05. (E) Serum starved GFP-EhRho5 trophozoites were stimulated in the presence as well in absence of 15μM LPA followed by mounting. Images were acquired using a confocal microscope (Scale Bar = 10μm) (F) Quantitative analysis represents relative pixel intensities of GFP-EhRho5 cells in LPA treated and untreated cells. Values in the SuperPlot are represented as the mean ± SEM of three independent experiments (N = 3, n>90, unpaired Student's $t$-test *<0.05). (G) Dot plot shows the percentage of cells exhibiting membrane associated EhRho5 in presence as well as in absence of LPA, in GFP-EhRho5 trophozoites. Values are represented as mean ± SEM of three independent experiments (N = 3, n≥90; unpaired Student's $t$-test, *p<0.05). (H) Serum starved GFP-EhRho5 trophozoites were stimulated with LPA. Cells were analysed using the FRAP module for 1 min after photobleaching. Representative images are shown, before and at the time points indicated after photobleaching (Scale Bar = 10μm). (I-J) Fluorescence recovery curve and recovery time $\tau_{1/2}$ for indicated cells are obtained using double exponential fit. Data is plotted as mean ± SEM from three biological replicates (n≥13). Time constant was determined at the plasma membrane and endomembrane by plotting the experimental data in a semi logarithmic scale, and regression analysis was performed. Significance was checked using unpaired Student's $t$-test, **<0.01.

stimulation increased the translocation rate of EhRho5 by 25.0% and 21.4% on membrane and vesicles, respectively (Fig 2I and 2J).

## LPA stimulation enhances macropinocytosis in EhRho5 dependent manner

Macropinocytosis is accompanied by major rearrangements in the actin cytoskeleton followed by membrane ruffling to engulf extracellular fluid [55]. These changes in cytoskeleton are largely governed by the Rho family of GTPases [56]. Therefore, we sought to ask if EhRho5 is involved during macropinocytosis in *E. histolytica*. Using trigger mediated RNAi silencing of EhRho5, we cloned EhRho5 in p4Trigger plasmid and generated 4Trigger-EhRho5 transgenic trophozoites. The silencing is mediated by Antisense sRNAs of an endogenously silenced gene (trigger), that are complementary to the gene fused to the trigger region [57]. The reduction in expression of the GTPase was measured by semi-quantitative PCR (S3A Fig). We then went ahead with measuring the dextran uptake efficacy of EhRho5 depleted and wild type trophozoites using Texas Red Dextran (TR-Dextran) as the fluid phase cargo [10]. Trophozoites were incubated with TR-Dextran for 15 mins in the presence and in the absence of LPA. Images obtained from confocal microscopy revealed 1.8-fold reduced dextran uptake in EhRho5 silenced amoebae, compared to wild-type trophozoites suggesting that EhRho5 plays an important role during constitutive macropinocytosis in *E. histolytica*. Upon LPA stimulation, we observed that WT trophozoites exhibited 1.2-fold increase in dextran uptake, while EhRho5 silenced trophozoites did not show any change in macropinocytic efficiency, indicating that EhRho5 is also involved during LPA stimulated macropinocytosis (Fig 3A and 3B). Moreover, we also observed slower proliferation of EhRho5 depleted trophozoites, which corroborates with previous reports stating that amoeboid organisms depend on macropinocytosis for their nutrient uptake, required for the sustenance (S3B Fig) [58].

For further confirmation on involvement of EhRho5 in macropinocytosis, we also performed dextran uptake assay in HA-EhRho5, HA-EhRho5CA and EhExHA (Empty vector) overexpressing trophozoites. While HA-EhRho5 trophozoites showed no significant change in the macropinocytic efficiency, the constitutively active mutant demonstrated 1.5-fold increased dextran uptake efficiency (Fig 3C and 3D). Live cell imaging of GFP-EhRho5 trophozoites revealed the participation of EhRho5 at macropinocytic cups and macropinosomes (S4 Video). Although macropinocytosis is a constitutive phenomenon, it is also known to be regulated by growth factors [59]. EGF stimulation has shown to induce macropinocytosis via

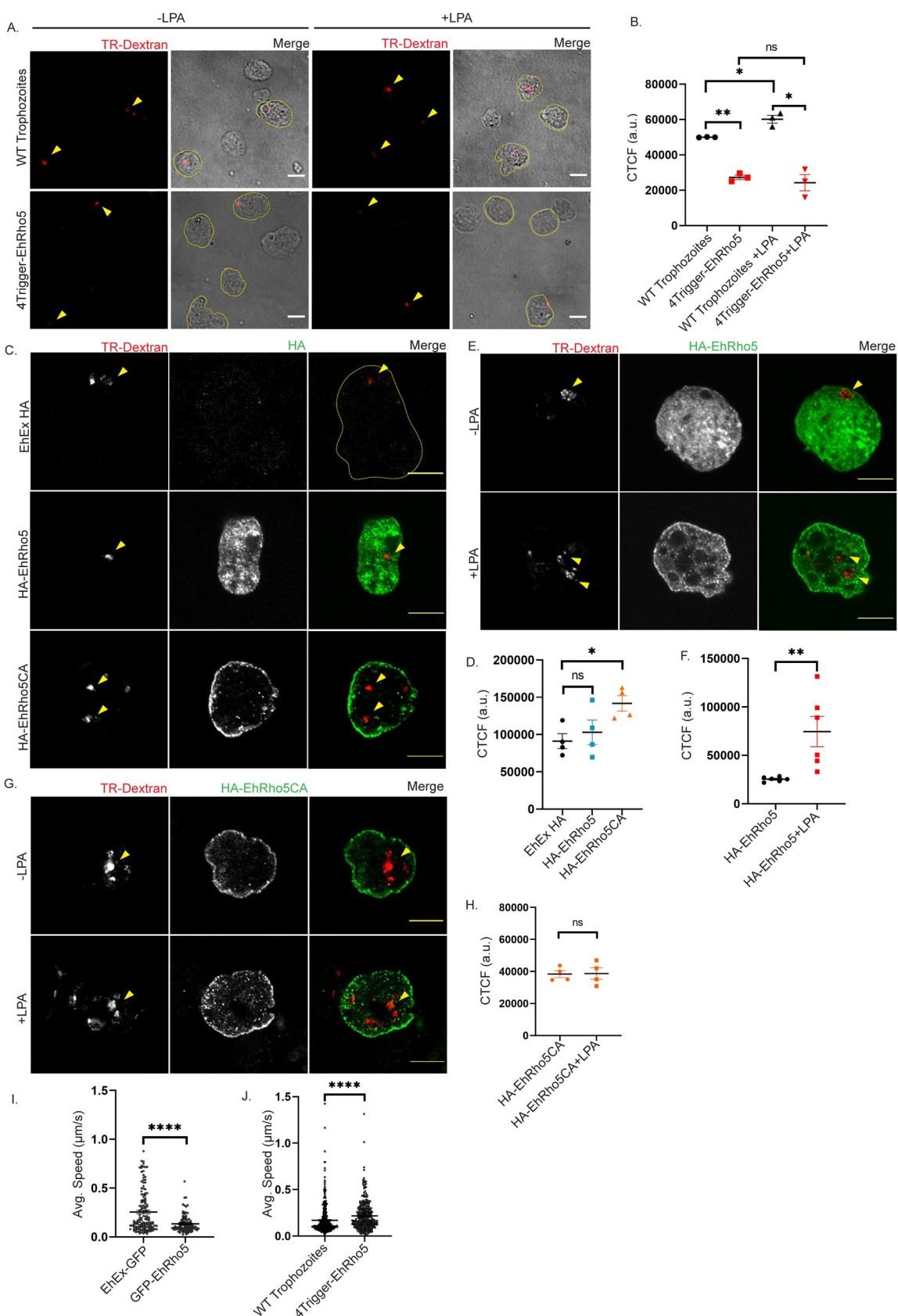

**Fig 3. EhRho5 is involved in LPA stimulated macropinocytosis.** (A) EhRho5 depleted and wild type trophozoites were serum starved and incubated with TR-Dextran (100μg/ml) in presence as well as in absence of LPA (15μM). Cells were fixed and processed for imaging. Arrowheads indicate the dextran within cells (Scale bar = 10μm). Cell boundary represents the trophozoites with dextran. (B) CTCF (Corrected Total Cell Fluorescence) was quantified as described in Materials and Methods, across indicated conditions and plotted (N = 3, n>150). Values represent mean ± SEM. Ratio paired Student's $t$-test, *p<0.05, **p<0.01, non-significant (ns) p>0.05. (C) Indicated cell lines were incubated with TR-Dextran followed by fixation and immunostaining with HA-antibody. Representative images show cells with internalised dextran (Scale bar = 10μm). Arrowheads indicate the Dextran within cells. Cell boundary is represented in a yellow dotted line for EhEx HA trophozoites (Empty vector). (D) CTCF was quantified and plotted showing means ± SEM (N = 4, n = 50). Ratio paired Student's $t$-test, *p< 0.05, non-significant (ns) p>0.05. (E-F) HA-EhRho5 trophozoites were incubated with TR-Dextran in presence as well as in absence of LPA. Cells were fixed, immunostained with HA-antibody, followed by image acquisition (Scale bar = 10μm). CTCF was calculated from images and plotted as mean ± SEM (N = 6, n>200). Ratio paired Student's $t$-test, **p< 0.01. Arrowheads represent the dextran in the cells. (G-H) HA-EhRho5CA trophozoites were incubated with Dextran (Scale bar = 10μm). Arrowheads indicate the Dextran within cells. CTCF was calculated and values were plotted in the graph as means ± SEM (N = 4, n>150). Ratio paired Student's $t$-test, non-significant (ns) p>0.05. (I-J) Trophozoites of indicated cell lines were incubated in a glass bottom petridish for 30mins in BI media. Live videos were acquired and their migration was studied. Average speed was calculated using ICY software and plotted. (N = 3, n>140, unpaired Student's $t$-test, p****<0.0001).

activation of Rac [19,60]. Therefore, HA-EhRho5 trophozoites were induced with LPA to activate EhRho5, along with TR-Dextran to assess their macropinocytic intake. We observed that macropinocytic efficiency of the trophozoites increased by 2.9-fold, upon LPA stimulation (Fig 3E and 3F). Next, we wanted to check if LPA stimulation can further increase the dextran uptake efficiency of EhRho5 constitutively active mutant. HA-EhRho5CA trophozoites were incubated with TR-Dextran for 15 mins both in the presence and in the absence of LPA and processed for imaging. LPA stimulation exhibited no change in macropinocytic efficiency of the constitutively active GTPase mutant (Fig 3G and 3H). Collectively, we confirmed the importance of EhRho5 during constitutive and LPA mediated macropinocytosis.

Formation of macropinosomes is initiated by membrane ruffles, which are largely defined as a patch enriched in phospholipids and small G-proteins [55]. These ruffles can also extend as a pseudopod, relevant to cell migration [26]. Membrane ruffles for both, macropinocytosis as well as pseudopod formation are actin dependent processes. A negative correlation between macropinocytosis and cell migration has been described for immune cells and *Dictyostelium* [61,62]. To study the role of EhRho5 in random migration, videos of EhEx-GFP, GFP-EhRho5, EhRho5 depleted (4Trigger-EhRho5) transgenic trophozoites along with WT trophozoites were acquired using live cell confocal microscopy. WT and EhRho5 depleted trophozoites were labelled with a fluorescent cell tracker for visualisation. Average speed of the cells was quantified using a programme on the Icy platform [63]. The trophozoites overexpressing GFP-EhRho5 showed more roundness and migrated slower than the GFP expressing trophozoites (Figs 3I and S3C). Their incapacity to deform and extend a pseudopod allowing migration is illustrated in the supplementary video (S5 Video). On the contrary, EhRho5 depleted trophozoites showed relatively higher speed, and subtle but non-significant difference in morphology (Figs 3J and S3D). The capacity of the cell to deform largely with extension of a pseudopod at the leading edge, in the migration direction is comparable between the wild type and the EhRho5 depleted trophozoites (S6 Video). The above results imply that EhRho5 is shared as a common machinery during macropinocytosis and migration in *E. histolytica*.

## LPA mediated EhRho5 activation occurs via PI Kinases

PI3K acts as a key player maintaining the levels of phosphatidylinositol phosphates during macropinocytosis in amoeboid cells [25,64] and accordingly, inhibition of PI3K led to reduced macropinocytosis [18,55]. Therefore, to decipher the involvement of PI3K in LPA stimulated signalling in *E. histolytica*, we utilized wortmannin- a cell permeable inhibitor of PI3Ks and PI4Ks [65]. Serum starved HA-EhRho5 trophozoites were treated with wortmannin, post LPA

stimulation and the localisation of the GTPase was studied. We observed that upon wortmannin treatment, the translocation of HA-EhRho5 to the plasma membrane as well as endomembrane reduced by 41.5%, while DMSO treatment showed no difference (Fig 4A and 4B). Similar observations were drawn when GFP-EhRho5 trophozoites were treated with wortmannin to investigate for translocation of the GTPase. (S4A Fig). This result confirmed that PIK inhibition abrogates the translocation of EhRho5. We further investigated if wortmannin blocks LPA induced EhRho5 activation, using effector pulldown assay as employed in Fig 2C and 2D. We observed that PIK inhibition leads to a marked reduction in RBD bound EhRho5 levels, compared to controls (Figs 4C and 4D and S4B). Therefore, we could establish that EhRho5 activation is also hampered by wortmannin based inhibition. Similarly, we determined the involvement of PIK in LPA mediated macropinocytosis. As observed earlier, LPA stimulation led to increased macropinocytic intake by HA-EhRho5 trophozoites (Fig 3E and 3F). However, upon wortmannin treatment, 1.7-fold reduction in dextran uptake efficiency was observed (S4C Fig), in line with the previous reports [18,66,67].

After assessing the effect of wortmannin on decrease in active EhRho5 levels, we next sought to determine the change in EhRho5 dynamics on wortmannin treatment with FRAP studies as described in the previous section (Fig 2H–2J). LPA stimulated GFP-EhRho5 overexpressing cells were treated with wortmannin and incubated for 15 mins. A part of the plasma membrane or a vesicular membrane was photobleached and the recovery was monitored. In a similar manner, the recovery of fluorescence was fitted using a double exponential model. PIK inhibition led to an increase in GFP-EhRho5 recovery time at plasma membrane with a $\tau_{1/2}$ of 6.21±0.8 sec, compared to control trophozoites with $\tau_{1/2}$ of 3.6±0.1 sec. At Vesicles, wortmannin treatment led to slower recovery compared to control, exhibiting a $\tau_{1/2}$ of 9.4 ±0.9 sec and 3.6 ±0.3 sec, respectively. Using quantitative analysis for $\tau_1$, we found that wortmannin treatment substantially increased the recovery time by 42.0% and 61.0% for both, plasma membrane as well as vesicular membrane, respectively (Fig 4E–4G and S7 and S8 Videos and S2 Table). Also, we observed no change in $\tau_2$ upon PIK inhibition (S4D Fig). Collectively, we could state that during macropinocytosis, LPA stimulated translocation and activation of EhRho5 requires PI Kinase activity. Also, reduced levels of active EhRho5 post wortmannin treatment indicate that the PIK lies upstream of EhRho5 (Fig 4C).

## EhGEF2- an *in vitro* EhRho5 GEF, is involved in LPA stimulated macropinocytosis

The cycling of GTPase from GDP to GTP bound form is invigilated by RhoGEF. Earlier, it has been demonstrated that phosphatidylinositol triphosphate (PtdIns(3,4,5)P₃) mediated activation of Rac involves RhoGEF [68]. We hypothesised that the activation of EhRho5 by LPA could be mediated by a guanine nucleotide exchange factor (GEF). Earlier studies based on biochemical and biophysical approaches, reported EhGEF2 and EhFP4 as potential GEF candidates for EhRho5 [39,69]. Both these candidates were selected for the current study. Nucleotide sequences corresponding to the RhoGEF domain of *Ehfp4* and *Ehgef2* genes were codon optimised and cloned in bacterial expression vector pET28a+ (GenScript). The recombinant plasmids were transformed into BL21(DE3) cells and proteins were expressed and purified using affinity chromatography (S5A and S5B Fig). We compared the *in vitro* GEF activities of both the candidates in order to identify the GEF. Firstly, the nucleotide binding capacity of the GTPase was assessed using fluorescence-based kinetics as described in Materials and Methods. EhRho5 protein was functional and exhibited faster exchange in presence of EDTA in the reaction buffer (S5C Fig). To identify the GEF, we utilised the same approach and studied the kinetics of exchange in the presence and absence of candidate GEFs. Our findings suggested

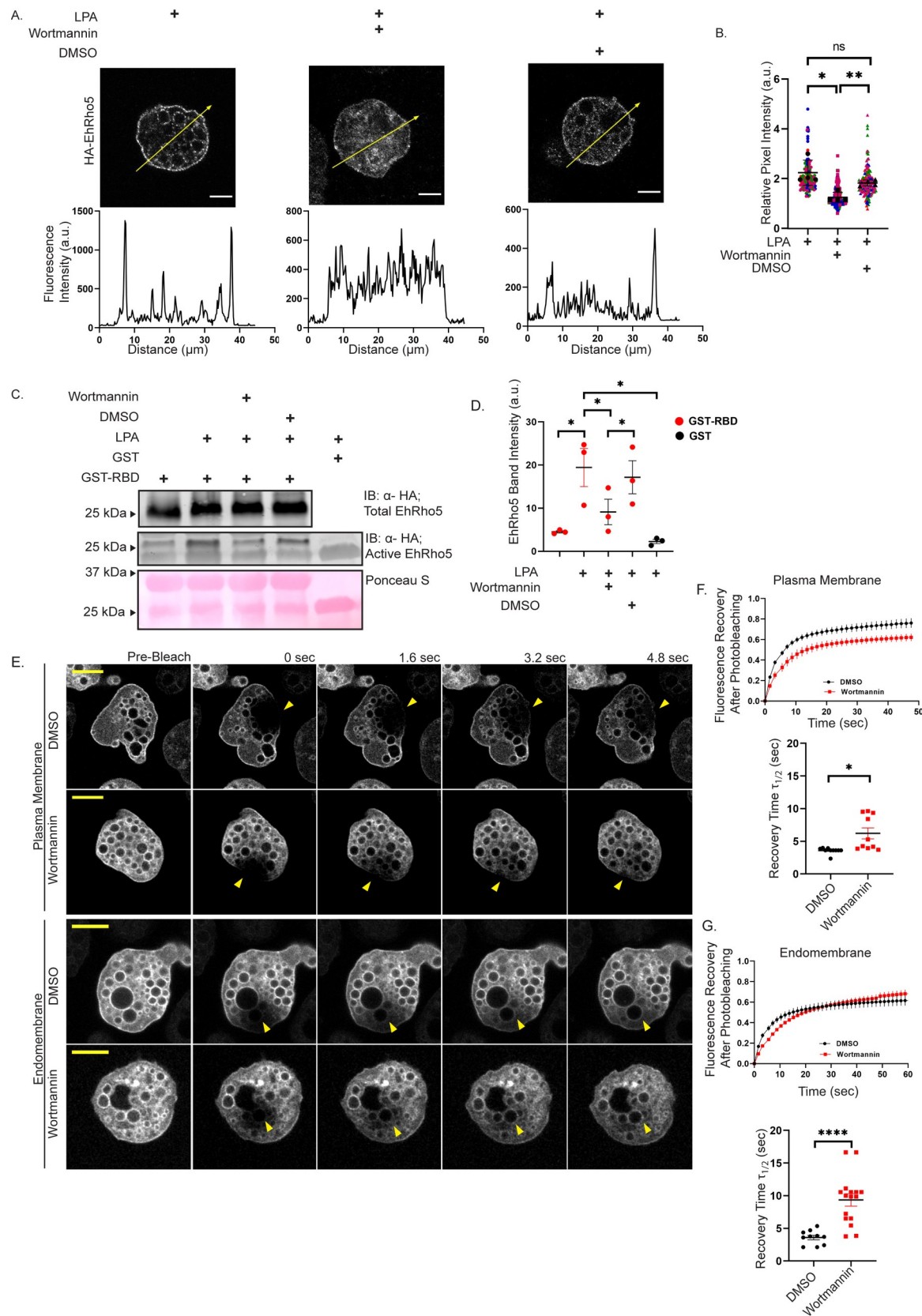

**Fig 4. PI Kinase inhibition abrogates LPA mediated activation of EhRho5.** (A) Immunofluorescence of HA-EhRho5 cells after treatment with wortmannin in LPA pre-treated condition. Serum starved trophozoites were stimulated with LPA (15μM) followed by treatment with wortmannin(100nM) or DMSO. Cells were processed, and imaged using a confocal microscope (Scale bar = 10μm). Distribution of fluorescence intensity over the arrow is shown in the line intensity plot. (B) Quantitative analysis of the relative pixel intensities upon wortmannin treatment in LPA pre-treated cells. A patch of the same area for plasma membrane, endomembrane and cytosol was selected and fluorescent intensity was determined. Calculation of relative pixel intensity was done by normalisation of plasma membrane and endomembrane intensity to that of cytosol for each cell. Relative pixel intensities were compared for different conditions. (N = 3, n>120; mean ± SEM; unpaired two-tailed Student's $t$-test, **p<0.01, *p< 0.05). (C) Serum starved HA-EhRho5 trophozoites were treated with LPA, followed by wortmannin. Cells were lysed and active EhRho5 pool was determined as described in Fig 2C and 2D. immunoblotted with HA-antibody. Total EhRho5 was used as loading control and DMSO as vehicle control. (D) Quantification of active EhRho5 population across various treatments was performed. Band intensity of active EhRho5 was background subtracted and normalised with bait (RBD/GST). Panel represents mean ± SEM of the band intensities of indicated treatments across three independent experiments (N = 3, ratio paired Student's $t$-test; *p<0.05). (E) GFP-EhRho5 trophozoites were incubated with wortmannin (100nM) or DMSO in LPA pre-treated condition, analysed using FRAP for 1 min after photobleaching. Representative images before and at the time points indicated after photobleaching are shown. Arrowheads indicate the photobleaching site (Scale bar = 10μm). (F-G) Fluorescence recovery curve of FRAP analysis and comparison of $\tau_{1/2}$ for indicated cells are shown, respectively. Values represents mean ± SEM across three independent experiments (n≥10, unpaired Student's $t$-test *p<0.05, ****p<0.0001).

that EhGEF2 enhanced the rate of nucleotide exchange (Fig 5A). EhRho1 (EHI_013260), a close homologue of EhRho5, harbouring ~93% identity exhibited slower exchange in the presence of EhGEF2 compared to EhRho5 (S5D Fig). Further, we studied the dose dependence of EhGEF2 activity on EhRho5 to determine the $K_{obs}$ (Fig 5B). Catalytic efficiency was obtained from the slope of linear least square fit of the $k_{obs}$ values against the concentrations of EhGEF2. We observed enhanced GEF activity as the concentration was increased, confirming EhGEF2 as the *in vitro* exchange factor for EhRho5 with a catalytic efficiency of 0.1 $M^{-1} S^{-1}$ (Fig 5C). EhRacG has been previously reported as a preferential substrate of EhGEF2 over EhRho5, therefore we also examined the catalytic efficiency of EhGEF2 for EhRacG. We observed that EhGEF2 acted as a GEF for EhRacG, with a catalytic efficiency of 0.7 $M^{-1} S^{-1}$, which is comparable to the efficiency observed for EhRho5 (S5E and S5F Fig).

Next, we wanted to determine the functional importance of EhGEF2 in trophozoites. Our previous results have established that EhRho5 is a substrate of EhGEF2 *in vitro* and is involved in macropinocytosis. Therefore, we sought to check the dextran uptake efficiency of EhGEF2 in trophozoites, to examine if it functionally phenocopies the substrate. Transgenic trophozoites for GEF depletion were generated using trigger mediated RNAi based silencing and depletion of EhGEF2 was confirmed with semi-Q PCR (S5G Fig). Serum starved EhGEF2 depleted trophozoites were incubated with TR-Dextran, in presence as well as in absence of LPA, and compared to wild type trophozoites for assessment of their macropinocytic efficiencies. We observed a 1.7-fold reduction in dextran intake of EhGEF2 depleted trophozoites (Fig 5D and 5E). EhGEF2 depleted trophozoites exhibited no alterations in their macropinocytic efficiency upon LPA stimulation. Our results suggest that similar to its substrate, EhGEF2 is crucial for constitutive and LPA stimulated macropinocytosis.

Previously, it has been reported that PI3K mediated activation of Rac GTPase involves translocation of GEFs to plasma membrane [70]. We hypothesised that EhGEF2 may translocate towards the plasma membrane upon experiencing stimuli. Therefore, we made transgenic trophozoites overexpressing HA-EhGEF2. The expression and localisation of the protein was confirmed using Western blotting and immunofluorescence, respectively (S5H and S5I Fig). We stimulated serum starved HA-EhGEF2 trophozoites with LPA and examined EhGEF2 localisation. HA-EhGEF2 translocated to membrane periphery upon LPA stimulation (Figs 5F and 5G and S5J and S9 Video). To probe the involvement of PI Kinase in translocation of EhGEF2, we used wortmannin based inhibition of PIKs. Serum starved HA-EhGEF2 trophozoites were treated with wortmannin post LPA stimulation and fixed with paraformaldehyde. Cells were then processed for immunofluorescence and images were acquired using a confocal

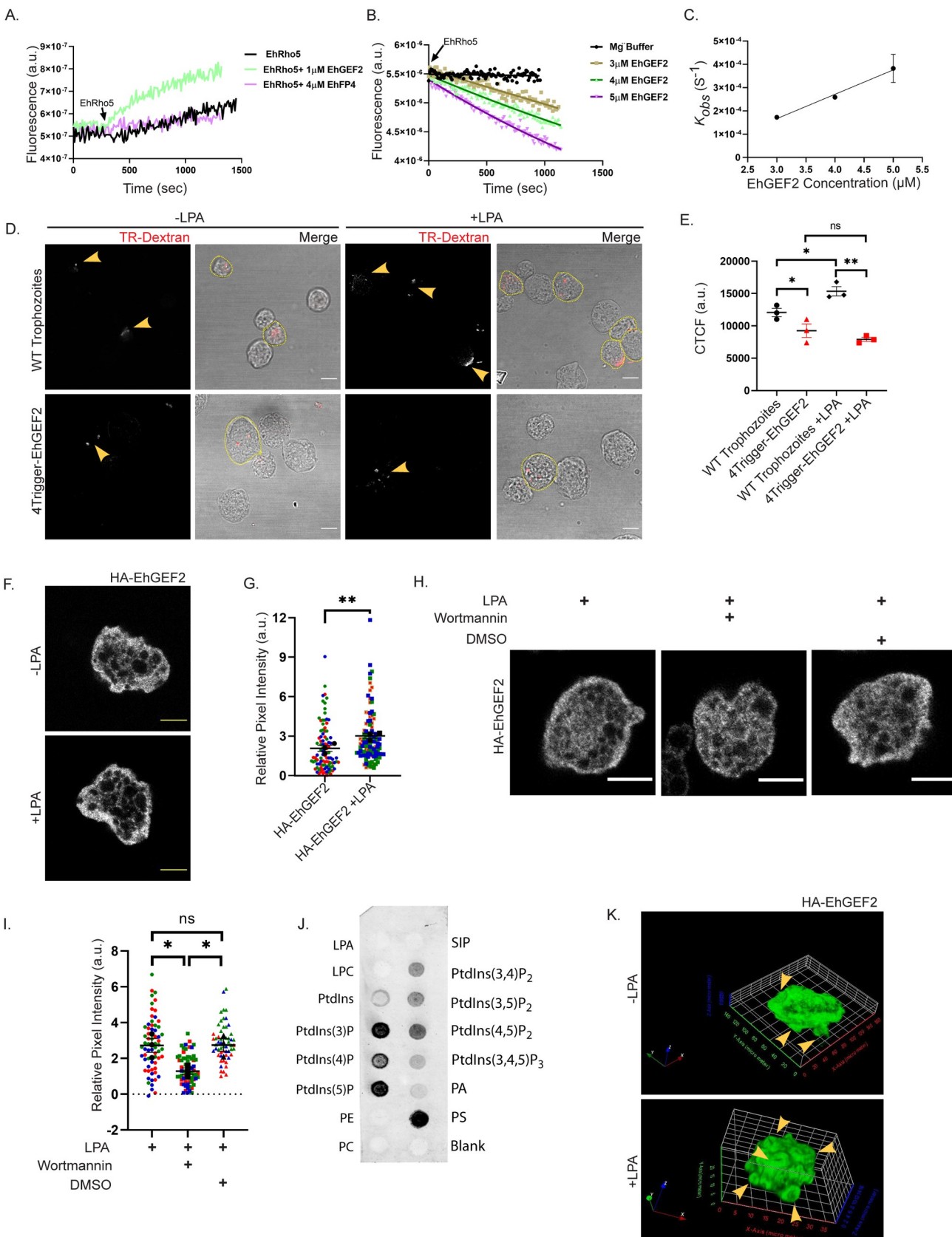

**Fig 5. EhGEF2, an EhRho5 GEF is involved in LPA stimulated macropinocytosis.** (A) Nucleotide exchange kinetics are shown for EhGEF2 (1μM) as well as EhFP4 (4μM) with EhRho5. Arrow indicates the addition of GEF during the reaction. (B) GEF activity exhibited by different concentrations of EhGEF2 for 2μM of EhRho5. Panel shows a single replicate data trace. (C) Catalytic efficiency ($k_{cat}/k_m$) was obtained from the slope of a linear least square fit of $k_{obs}$ values against EhGEF2 concentration from three independent datasets. Values represent mean ± SEM of three independent measurements (N = 3) (D) EhGEF2 and Wild-type (WT) trophozoites were incubated with TR-Dextran in the presence or in the absence of LPA (15μM) for 15 mins at 37˚C and fixed with PFA. Cells were analysed in Olympus FV3000 confocal microscope. Arrows indicate TR-Dextran within the cells. Cell boundary highlights the cell with dextran (N = 3, n≥150; scale bar = 10μm). (E) Quantification of Dextran uptake in indicated trophozoites by calculating CTCF. The graph represents mean ± SEM, ratio-paired Student's $t$-test; * $p<0.05$; ** $<0.01$; non-significant (ns), $p>0.05$. (F) HA-EhGEF2 trophozoites were stimulated in presence as well as in absence of LPA (15μM). Cells were then fixed and incubated with anti-HA antibody, followed by Alexa labelled secondary antibody. Panel illustrates the translocation of EhGEF2 towards the membrane periphery, post LPA treatment (Scale bar = 10μm). (G) Quantification of relative pixel intensities near the membrane in EhGEF2 trophozoites ±LPA (15μM). Statistical significance was determined using unpaired Student's $t$-test (N = 3, n>100; **$p<0.01$). (H) Serum starved trophozoites were stimulated with LPA followed by wortmannin treatment. Cells were fixed and proceeded for immunofluorescence using anti-HA antibody (Scale bar = 10μm). (I) Quantification of relative pixel intensities across indicated conditions. Statistical significance are determined using unpaired Student's $t$-test (N = 3, n≥65; * $p<0.05$) (J) Nitrocellulose membrane with different lipids spots was blocked with 5% BSA prepared in 1X TBST and then incubated with EhGEF2 at 4˚C overnight. The binding of EhGEF2 was detected using anti-His antibody. (K) Representative 3D reconstruction of HA-EhGEF2 trophozoites in the presence and in absence of LPA. Arrowheads indicate the macropinocytic cups on the surface of the trophozoites.

microscope (Fig 5H). Inhibition of PIKs led to reduction in translocation of HA-EhGEF2 at the plasma membrane periphery by 52.6%, whereas DMSO treatment showed no significant changes (Fig 5I). These results confirmed the existence of PI Kinase upstream of EhGEF2.

The above results confirmed the involvement of PIK activity in EhGEF2 translocation and prompted us to hypothesise that EhGEF2 should recognise a phosphorylated form of membrane PI. Thus, we next sought to identify the lipid binding specificity of EhGEF2. Recombinant His-EhGEF2 protein was employed to determine the lipid binding in a lipid overlay assay (Figs 5J and S5K). The results showed that EhGEF2 could bind strongly to PI(3)P, PI(4)P, PI(5)P and phosphatidylserine, along with relatively weaker binding to PtdIns(3,4)P2, PtdIns(3,5)P2, PtdIns(4,5)P2 and PtdIns(3,4,5)P3. In addition, EhGEF2 also recognised PA with variable binding capacity over different biological replicates (S5K Fig). The results from the *in vitro* lipid binding assay, thus corroborated with our wortmannin based inhibition studies (Fig 5H and 5I), suggesting an involvement of PIKs in LPA mediated EhGEF2 translocation.

Overexpression of HA-EhGEF2 also led to formation of several crown-shaped membrane invaginations resembling macropinocytic cups. The presence of F-actin in these structures is demonstrated by Alexa-568 labelled phalloidin staining (S5L Fig). Interestingly, EhGEF2 also localised at the tips of these invaginations while engulfing dextran (S5M Fig). LPA stimulation further enhanced the formation of these structures in HA-EhGEF2 overexpressing trophozoites (Fig 5K). Altogether, we conclude that EhGEF2 acts as *in vitro* GEF for EhRho5 and possibly regulates activation of EhRho5 during macropinocytosis in PI Kinase dependent manner.

## Discussion

Amoebic trophozoites utilize a variety of cellular processes for sustenance of pathogenesis. Majority of these processes are accompanied by dynamic changes in cytoskeleton, which are regulated by Rho family of GTPases [25,71]. As a member of Ras superfamily of small GTPases, Rho GTPases shuttle between an inactive GDP bound state and an active GTP bound state; controlled by regulators such as GEFs, GAPs and GDIs. Translocation of these GTPases to membrane is known to be associated with their activation [46,49]. Active Rho GTPases interact with effector proteins to relay numerous downstream signals [48]. Extracellular growth factors are known to activate Rho GTPases via associated GEFs and PI3K axis [19,30,72]. In the current study, we have examined the components that govern constitutive and LPA stimulated macropinocytosis in *E. histolytica*.

Using confocal microscopy and biochemical approaches we established that LPA stimulates the translocation and activation of EhRho5 (Figs 1 and 2). LPA mediated signalling has been

implicated to function via GPCRs, to elicit diverse biological responses across various mammalian cell lines [73]. Till date, three distinct GPCRs have been reported to contribute to LPA signalling in different tissues [29] Moreover, RhoA and Rac activation has been linked with LPA signalling via LPA$_1$ receptors encoded by *edg2* gene [74]. Although a GPCR has been reported which directs bacterial engulfment in response to LPS (Lipopolysaccharide) stimulation, LPA receptors are yet to be reported in *E. histolytica* [75].

Rho GTPases interact with the effectors only in their GTP bound form and this property has been thoroughly exploited in the field to identify activated pool of the GTPases using effector pulldown assay [32,53] The effector binding by Rho GTPases is generally attributed to Switch I region also known as the effector binding region (YVPTVFDNY; EhRho5) and the 'Rho insert' region (EAMIRKLADENQK; EhRho5) [76]. Among these, the switch I region is in close vicinity to the nucleotide moiety and responsible for sensing GTP/GDP through interaction with the Mg$^{2+}$ ion via a conserved threonine (Thr42; EhRho5). Although the above regions are crucial for binding to the effector molecules and are conserved among the different Rho family members, less conserved regions across the G-domain of GTPases too determine effector binding specificity [77,78]. Thus, although the preferential binding to RBD by EhRho5 (Figs 2A and 2B and S2D) may be attributed to the distinct regions on the GTPase including Switch I and the Rho insert helix, the molecular explanation for differential affinity of EhRho5 for RBD and PBD would require the 3dimensional structural analysis of RBD or PBD bound complexes of EhRho5. We have demonstrated nucleotide dependency in EhRho5 binding to RBD (Figs 2A and 2B and S2D) and concluded that the GTPase belongs to Rho subfamily. Using the same pulldown approach, we established that LPA stimulation leads to increase in active EhRho5 levels in the trophozoites (Figs 2C and 2D and S2E).

Traditionally, Ras super family small GTPases are known to be activated by GEFs [79]. Our biophysical and cell-based studies, together demonstrated EhGEF2 as an exchange factor for EhRho5 (Fig 5). Previously, EhGEF2 has been shown to activate EhRacG as its preferred substrate among a set of Rho family GTPases inclusive of EhRho5, but we observed a comparable catalytic efficiency of EhGEF2 for EhRacG and EhRho5 (Figs 5B and 5C and S5E and S5F) [69]. Nevertheless, though the *in vitro* catalytic efficiency provides mechanistic information about the substrate specificity of a GEF at the molecular level, the cellular activity of a GEF on a GTPase also largely relies on the relative sub-cellular localization of the GEF-substrate pair. While under normal growth conditions, EhRacG, similar to EhRho5 has been shown to be predominantly localised in cytosol, its localization upon LPA stimulation is yet to be studied [80]. Therefore, the present information seems inadequate to establish whether EhGEF2 has promiscuous activity on both the *in vitro* GTPase substrates. Of note, promiscuity of guanine exchange factors (GEF) or GTPase activating proteins (GAP) are not rare in the literature [81,82].

EhGEF2, as its substrate GTPase EhRho5, translocated towards the periphery of the plasma membrane upon LPA treatment (Figs 1A and 1B and 5F and 5G). Earlier studies have shown how activity and specificity of RhoGEFs is regulated by various mechanisms including subcellular sequestration upon experiencing a physiological stimulus [69,70,83–85]. For example, redistribution of Tiam1 upon PDGF or LPA stimulation from cytoplasm to plasma membrane leads to activation of human Rac1 [70]. Growth factors like PDGF and insulin stimulation have been reported to enhance PIP3 levels in the cells, required for activation of Rac GTPase [86]. We investigated the role of PI-Kinase in LPA mediated translocation and activation of EhRho5. Our results are consistent with previous reports showing that PI-Kinase is required for growth factor mediated activation of GTPase (Fig 4) [86]. We observed spatio-temporal regulation of EhGEF2-EhRho5 flux upon LPA stimulation (Figs 1 and 5F and 5G), where the PIK dependent recruitment and the activation of the GTPase relies on the presence of EhGEF2

at the plasma membrane. These results led us to hypothesise that EhGEF2 might recognise wortmannin sensitive PtdIns. In contrast to previously reported study [69], we demonstrated that the EhGEF2 indeed recognises PIs, such as PI(3)P, PI(4)P, PI(5)P, PtdIns(3,4)P2, PtdIns (3,5)P2, PtdIns(4,5)P2 and PtdIns(3,4,5)P3 along with phosphatidic acid and phosphatidylserine (Figs 5J and S5K). It is known that the generation of PI(3)P, PI(3,4)$P_2$, PI(3,4,5)$P_3$ and PI (4)P is governed by wortmannin sensitive PI3K and PI4K family of lipid kinases [87]. In *Entamoeba histolytica*, while PtdIns(3,4,5)P3 localized on the extended pseudopodia and phagocytic cups [88], PtdIns3P on the phagocytic cups, and internal vesicles [89], PI(4)P and PI(3,4) $P_2$ binding proteins were shown to reside on the plasma membrane [57,90]. Together our data suggests that membrane association of EhGEF2 is attributable to wortmannin sensitive signalling of different phosphorylated forms of phosphatidylinositol.

Rho family members are known to translocate and activate signalling upon experiencing physiological stimuli [15,30,45,46,48]. The dynamic changes in flux of active GTPase create spatial signalling patterns. These spatial patterns give rise to what is known as activity zones and their formation is attributed to the GTPase cycle. Active zones are partially shaped by the machinery such as GEFs and GAPs, required for the activation of the GTPase [60,72,91]. Previous studies have reported the dynamic local activation and inactivation of RhoA and Cdc42, with half-lives of ~8–12 sec [92]. Here, we utilized FRAP based studies to demonstrate that EhRho5 takes 3–4 seconds to cycle from inactive cytosolic pool to active membrane bound pool, upon LPA stimulation (Fig 2H–2J). Our FRAP data was approximated by a double exponential model with fast and slow time constants $\tau_1$ and $\tau_2$ respectively. Our results indicate the existence of two independent processes recruiting EhRho5 at the plasma membrane. The time constant $\tau_1$ made significantly more contributions (recovery of fluorescence and amplitude) and responded to LPA thus representing the faster mode of EhRho5 recruitment. We hypothesised that the faster recovery rate $\tau_1$, signifies recruitment of the LPA stimulated, GEF catalysed EhRho5 pool, since GEF catalyses the GDP-to -GTP exchange and makes it faster (Fig 2H and 2I) [19,59,93,94]. We believe that the slower mode of EhRho5 recruitment, might be either vesicle fusion or through a slow process such as lateral diffusion [95,96]. This hypothesis is consistent with the fact that $\tau_1$ majorly contributes in the fluorescence recovery compared to $\tau_2$ (S1 and S2 Tables). The different $\tau_1$ for the recovery of EhRho5 at plasma membrane and endomembrane, probably is a result of varying concentrations of GEF or EhRho5, within the cell [15,97]. We further excavated into the dynamicity of this signalling and found that wortmannin based inhibition of PIK led to an increase in the recovery time $\tau_1$ of EhRho5 at both plasma membrane and endomembrane. Abrogation of PIK had no effect on $\tau_2$, suggesting that the slower mode of EhRho5 recovery may not be wortmannin sensitive (Fig 4). This property to exchange its cytosolic inactive pool to active membrane pool requires GTPase cycle. Previous reports have shown a rapidly exchangeable pool for Cdc42 and Rac1 on polar caps and phagosomal membrane [98,99]. This feature of EhRho5 can be compared to various small GTPases like Rho and Rab families where the cycling dynamics are controlled by the regulatory proteins.

In the protist *Entamoeba histolytica*, macropinocytosis is an important process for nutrient uptake and sustenance in the host. Although macropinocytosis is a constitutive process, it has also been shown to be stimulated by growth factors [20,100]. Stimulation with EGF and PDGF causes increase in actin driven ruffling, ultimately leading to macropinocytosis [101]. While Rac1, RhoA and Cdc42 are involved in regulation of macropinosomes formation, RhoC, on the other hand, governs the entire process [72,91,102]. In *D. discoideum*, a soil dwelling social amoeba, DdRac1 and PIP3 have been shown to localise at macropinocytic cups [103]. Here, using cell-based studies, we demonstrated that EhGEF2 and EhRho5 are important in constitutive, and also govern LPA stimulated macropinocytosis (Figs 3, 5 and 6). Serum starvation of the trophozoites leads to minimal localisation of EhRho5 on the plasma membrane

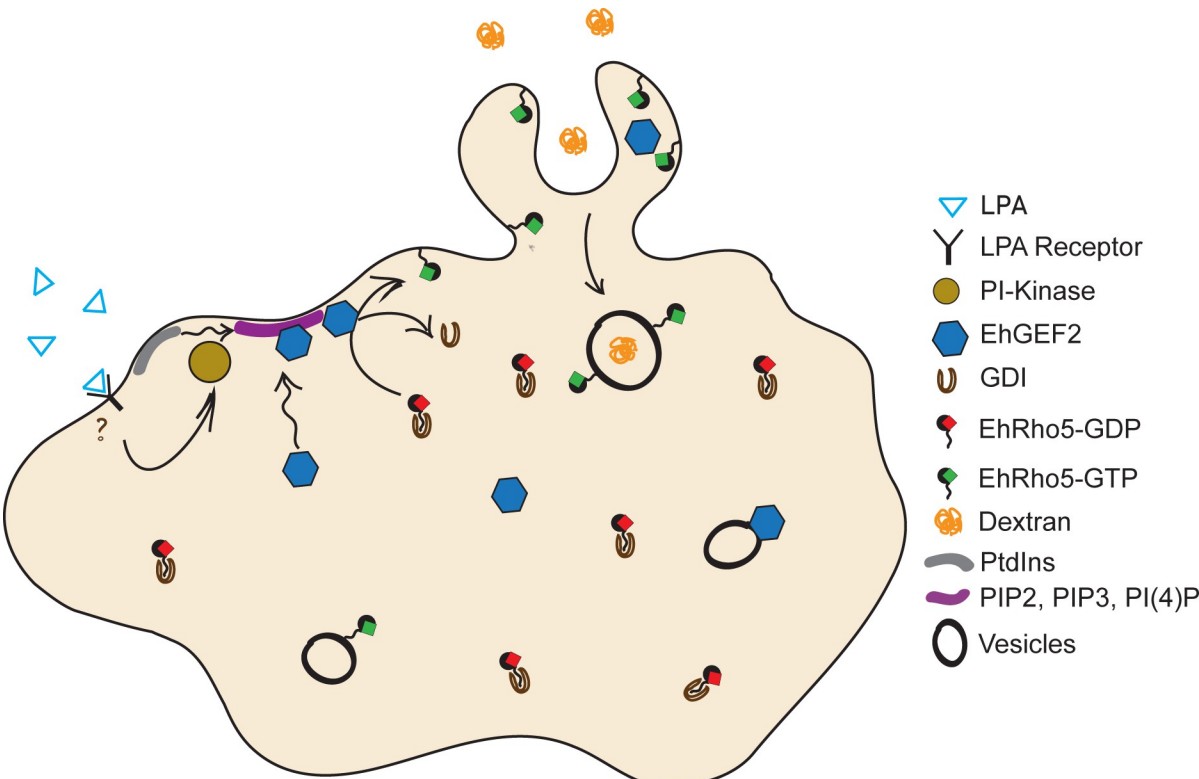

**Fig 6. A proposed model of LPA mediated macropinocytosis on PI Kinase-EhGEF2-EhRho5 axis.** Upon LPA stimulation, signals are relayed to PI Kinase. An increase in PtdIns leads to sequestration of EhGEF2 towards the periphery of the plasma membrane. As a result of high flux of EhGEF2 towards the membrane periphery, EhRho5 is activated and participates in macropinocytosis.

substantiating their reduced efficiency for macropinocytosis. EhGEF2 functionally phenocopies EhRho5, corroborating with its *in vitro* exchange activity on the GTPase (Figs 3, 5 and 6). However, overexpression of the GEF led to the formation of numerous macropinocytic cups at the surface, which became more pronounced post LPA stimulation (S5J Fig). The fact that no such phenotypic alterations were observed for HA-EhRho5 trophozoites, indicated the existence of more complex regulation of macropinocytosis, involving additional substrates GTPases for EhGEF2. In this context, EhRacG may also be a potential candidate to be investigated in future studies. Moreover, in line with the previous reports, we also observed a reduction in macropinocytic efficiency upon PIK inhibition in LPA pre-treated trophozoites (S4C Fig). These findings further strengthened our claim that EhGEF2 and EhRho5 represent downstream components in the LPA-PIK axis during macropinocytosis.

Often molecular machinery are shared between different processes but the way how the cell finds the right balance is not yet completely understood. Macropinocytosis and migration are known to be regulated by PIP3 levels within the cell, where low PIP3 levels shift the equilibrium towards migration [62,104]. Also, a recent report has shown that antigen capture, which is a form of phagocytosis, and migration are incompatible processes [61]. In the current study, we observed that while EhRho5 is crucial for fluid phase uptake, its overexpression abrogates amoebic migration, suggesting it's distinct role in these processes (Fig 3). Our findings suggest that macropinocytosis and migration might have EhRho5 as shared molecular machinery in *E. histolytica*. EhRho5 may participate in cytoskeleton dynamics to steer the switch between membrane extension for pseudopod or macropinocytosis.

In summary, the current study identified some of the crucial components of the machinery involved during constitutive and LPA stimulated macropinocytosis. Our results suggest the sequestration of EhGEF2 is possibly mediated by PIK for EhRho5 activation. It also sheds light on how their spatio-temporal dynamics contribute to macropinocytosis in *E. histolytica* (Fig 6).

## Materials and methods

### *E. histolytica* cultures

*E. histolytica* strain HM1:IMSS trophozoites were grown axenically in BI-S-33 medium supplemented with 15% (v/v) heat inactivated adult bovine serum (RM9981), 2.6% v/v vitamin mix 100U of penicillin/ml and 100 µg streptomycin sulphate/ml at 35.5˚ C. All the cell lines were maintained at 6 µg/ml of G418 concentration in BI-S-33 medium. For experiments, cell lines were maintained at 20 µg/ml of G418 and 30 µg/ml of tetracycline for 48 hrs in BI-S-33 medium, wherever necessary. Serum starvation for experiments was performed for a duration of 12hrs in BI medium. All the experiments were performed in BI medium (devoid of serum and vitamins), at least three times, unless otherwise mentioned.

### Cloning and plasmid construction

For generation of transgenic constructs, *EhRho5* (Acc no. EHI_012240), *EhGEF2* (Acc no. EHI_182740) were PCR amplified using fwd- cccgggATGTCAGCTGCACCAACAGATGC and rev-ctcgagTTACAACAAAGCACATTTCTTAGAAC and fwd- cccgggATGACAAAAGTAT TAGTTTCAC; rev-ctcgagTTAACTATTTGTAATTGAAGTTCTTTTTATTTT primers, respectively. The amplified fragment was further cloned into EhEx-HA, pTrigger (Kind gift from Dr. Upinder Singh, Stanford University) and pTEx-GFP (Tet- inducible plasmid; Kind gift from Dr. Tomoyoshi Nozaki) using SmaI- XhoI. To generate constitutively active and dominant negative mutant of EhRho5, site directed mutagenesis was done using fwd-GGAGCTGTTGGAAAAA ACTGTTTATTA and rev-ACAGTTTTTTCCAACAGCTCCATC, fwd-GTAGGAGATGTA GCTGTTGGAAAAACA and rev- ACATGTTTTTCCAACAGCTACATC, respectively. For construction of bacterial expression plasmids, the PCR fragments were cloned between EcoRI-X-hoI in pET28a+ under lac promoter and expressed in BL21 (DE3) cells. DNA sequences corresponding to RhoGEF domain of *EhFP4* (1–333 aa) and *EhGEF2* (390–732 aa) were codon optimised and cloned between NdeI-XhoI in pET28+ vector (http://www.genscript.com). All the constructs were sequence confirmed before generation of stable transfectants.

### Generation of stable transgenic trophozoites

To overexpress transgenic constructs, trophozoites were washed with phosphate buffer saline (1X PBS) followed by incomplete cytomix buffer (10 mM $K_2HPO_4/KH_2PO_4$ (pH 7.6), 120 mM KCl, 0.15 mM $CaCl_2$, 25 mM HEPES (pH 7.4), 2 mM EGTA, 5 mM $MgCl_2$). The washed cells are then re-suspended in 0.4 mL of complete cytomix buffer (incomplete cytomix containing 4 mM adenosine triphosphate, 10 mM reduced glutathione) containing 50–100 µg of plasmid DNA. Trophozoites were then subjected to pulse of 500 V voltage and 500 lF capacitance. Stable clones were selected in the presence of 4 µg/ml G418. Experiments were performed at 20µg/ml of G418 and 30 µg/ml of tetracycline additionally for tetracycline-inducible stable transfectants (GFP-EhRho5).

### Western blotting

Trophozoites were harvested and washed with 1X PBS, followed by lysis with lysis buffer (50 mM Tris-Cl pH 7.5, 150 mM NaCl, 1mM DTT and 1% NP-40) in the presence of

protease inhibitor mixture (100 μM leupeptin, 10 μM pepstatin A, 0.3 μM aprotinin, 1 μM PMSF, and 10 μM E-64). Proteins were resolved in 12–15% SDS-PAGE under reducing conditions. Then, proteins were transferred to the nitrocellulose membrane at 300mA for 3hrs (or 90mA for 16 hrs). The membrane was blocked in 5% (w/v) skimmed milk for 1 hr and probed with anti-HA (1:1000), anti-CS (1:1000) [105], anti-Hgl (1:40; 3F4 and 7F4) [106] primary antibodies overnight. Blots were washed with 1X PBST thrice and incubated with Alexa Fluor secondary antibodies for 1hr. Post washing with 1X PBST thrice, the membrane was dried and detected using Infrared detection system.

## Immunofluorescence assay

Serum starved cells were incubated on depression glass slide. Cells were stimulated with 15 μM LPA for 15 mins, followed by 100 nM wortmannin for 15 mins based on the requirement of the experiment. Cells were then fixed with 4%(w/v) paraformaldehyde at 37˚C for 15 min, followed by permeabilization with 0.1% triton X-100 (v/v). Blocking was done using 5% fetal bovine serum in 1X PBS (w/v). Then, cells were incubated with primary antibodies rabbit monoclonal anti-HA (catalogue number sc-7392, Santa Cruz Biotechnology), and anti-Hgl (1:100, 3F4 and 7F4) at room temperature [40]. After three washes in blocking solution, trophozoites were incubated with Alexa Fluor-conjugated (Life Technologies) secondary antibodies (1:500 dilutions) for 1 hr at room temperature. After three washes with blocking solution, coverslips were mounted on the glass slide using Mowiol. Slides were examined using a LSM-780 laser scanning confocal microscope (Carl Zeiss, GmbH, Jena, Germany) with a 63x/1.4 NA oil immersion objective lens.

Dextran uptake assay: Trophozoites were serum starved in BI media for 12 hrs at 35.5˚C before initiation of uptake assays. To follow dextran uptake in trophozoites, transgenic trophozoites were incubated with TR-Dextran (100μg/ml) in prewarmed BI media for 15 mins at 37˚C. After the incubation, cells were fixed and processed for imaging. Cells expressing the HA-tagged proteins were analysed for their dextran uptake, but in the depletion cell lines, all the cells were taken in account for assessing their dextran uptake efficiency. For analysis, z-stacks of the cell were combined to get maximum intensity projection (MIP) using ImageJ software [107]. A cell boundary was made and integrated intensity was calculated followed by subtraction of background intensity to obtain CTCF.

$$CTCF = Integrated\ density - (Area\ of\ selected\ cell \times Mean\ fluorescence\ of\ background\ readings)$$

## Analysis for translocation of EhRho5 and EhGEF2

Analysis was done using ImageJ software. An ROI of the membrane (A) and an adjacent cytosol (B), with identical area, was measured for the intensity in presence as well as in absence of LPA. The intensity of the plasma membrane was normalized to its cytosolic counterpart. The ratio was compared between LPA treated and untreated trophozoites.

For analyzing the peripheral loacalisation of EhGEF2, a cell boundary was made and intensity was determined for the cell 'X'. The boundary was then reduced by 1.5μm using enlarge command and the intensity was measured 'Y'. The difference of these intensities followed by normalisation with the total intensity of the cell (X) resulted in the relative pixel intensity which was compared over LPA treated and untreated condition.

$$Relative\ pixel\ intensity = \frac{X - Y}{X}$$

## Lipid overlay assay

The assay was performed using PIP strips according to the manufacturer's instructions (Thermo Scientific). The membrane was incubated with 1mg/ml of His-EhGEF2 in blocking buffer overnight at 4°C. The protein bound to lipids was detected by anti-His antibody.

## Live cell dextran uptake imaging

GFP-EhRho5 trophozoites were incubated on a glass bottom coverslip for 30–45 mins in BI media. TR-Dextran (2mg/ml) was added in the media and videos were acquired on Olympus FV3000 confocal microscope with a time interval of 1.6secs between each frame, for 200 frames.

## Live cell imaging to study random migration

Non-fluorescent trophozoites were labelled with 2µM cell tracker orange for 30 mins at 35.5°C. Cells were then washed with 1X PBS and incubated on a glass bottom coverslip for 30–45 mins. Videos were acquired on Olympus FV3000 confocal microscope with a time interval of 1.6 secs between each frame, for 200 frames.

## Image analysis of migration videos to quantify morpho-dynamic parameters

Analysis was performed using algorithms of the Icy software, a free and open-source platform for bioimage analysis that provides multiple resources to visualize annotate and quantify bioimaging data (http://icy.bioimageanalysis.org). Briefly, approximative polygonal Regions of Interest (ROIs) were drawn manually on frame 0 around each amoeba to initialize the segmentation for the Active Contours plugin (AC), then through AC method, each ROI polygon deforms to spouse the boundary of the segmented amoeba. Then, we used "Track objects over time" in the AC plugin with a volume constraint and volume weight 0.1. Once these parameters were established the segmentation was launched over the full sequence of frames. Once the segmentation was completed other the movie, AC sends the data to the Track Manager (TM) platform of the Icy software. As the amoeba shape was automatically tracked over time, the centroids of the successive ROIs were concatenated into a cell track by TM. The resulting tracks are directly overlayed on the original sequence. The TM module in Icy contains many Track Processors as Motion Profiler for quantifying cell motility parameters and cell morphology, such as cell speed (mm/s) and roundness (%), respectively. The speed is calculated with the displacement of the centroids of the successive ROIs over time and the "Roundness" is a measure of the similarity of the ROI to a circle.

## Recombinant protein purification

To express and purify RBD/PBD-GST (Kind gift from Dr. Richard Cerione) tagged recombinant proteins, their gene constructs were transformed into *E. coli* BL21(DE3) cells. A single expressing colony was inoculated in 1L LB broth with 100µg/ml ampicillin and the bacterial culture was grown at 37°C till the OD600 was 0.6. The cultures were then induced with 500µM IPTG and grown at 37°C for 3hrs. The cells were harvested and the pellet was lysed in lysis buffer with 50mM Tris pH 7.5, 200mM NaCl, 1 mM DTT, 1% Triton X-100, and 200 1mM PMSF. sonication was performed and centrifuged at 4°C, 18000 rpm for 30 mins. The cleared lysates were incubated with GST-sepharose beads for 1 hr at 4°C. The beads were washed thrice with wash buffer (50mM Tris pH 7.5, 100mM NaCl, 1 mM DTT and 200 1mM PMSF),

followed by washing with wash buffer containing glycerol. Bound proteins were eluted using 10mM glutathione. The proteins were concentrated using Amicon centrifugal filter unit.

To express and purify His (pET28a+) tagged recombinant proteins (EhRho5, EhRho1, EhRacG (EhRho2), EhGEF2, EhFP4) their gene constructs were transformed into *E. coli* BL21 (DE3) cells. A single expressing colony was inoculated in 1L LB broth with 30μg/ml kanamycin and the bacterial culture was grown at 37˚C till the OD600 was 0.6. The cultures were then induced with 200μM IPTG and grown at 18˚C overnight. The cells were harvested and the pellet was resuspended in lysis buffer with 50mM Tris pH 8.0, 300mM NaCl, 10mM Imidazole, 1% Triton X-100, 2mM $MgCl_2$ and 10mM β-mercaptoethanol, 200μM PMSF. The cells were lysed via sonication and cleared using centrifugation at 4˚C 18000 rpm for 30 mins. The cleared lysate was incubated with Ni-NTA beads for 20 mins on rotamer at 4˚C. The proteins bound non-specifically to the beads were washed off with wash buffer consisting 50mM imidazole. The protein was eluted with 200mM Imidazole and concentrated using Amicon centrifugal filter unit. All the purified proteins were evaluated for their purity using SDS-PAGE.

## Surface biotinylation assay

Membrane fraction was captured using surface biotinylation assay as described previously [108]. Briefly, serum starved HA-EhRho5 trophozoites were incubated with 15μM LPA and 0.25mg/ml EZ-link Sulfo-NHS-Biotin (Invitrogen, Cat. 21217) in 1X PBS (6.7 mM $NaHPO_4$, 3.3 mM $NaH_2PO_4$, and 140 mM NaCl, pH 7.2) at 4˚C for 30 min. To quench the unreacted biotin, cells were treated with quenching solution (50 mM Tris-HCl, pH 8.0). Post quenching, cells were washed with 1X PBS and lysed. For cell lysis we used NP40, a mild detergent to avoid complete dissolution of the plasma membrane (Buffer: 50 mM Tris-Cl pH 7.5, 150 mM NaCl, 1mM DTT and 1% NP-40 in the presence of protease inhibitor mixture-100 μM leupeptin, 10 μM pepstatin A, 0.3 μM aprotinin, 1 μM PMSF, and 10 μM E-64). Partial integrity of the lipid molecules in the membrane fragments thus generated, kept the integral and the tethered peripheral proteins associated. The lysate was incubated with NeutrAvidin beads for 30 mins at room temperature. The beads were washed with 1X PBS and the biotin labelled membrane fraction was eluted from the beads followed by Western blotting.

## Fluorescence recovery after photobleaching (FRAP)

FRAP was performed using the FRAP module on Olympus FV3000 confocal microscope. For FRAP, serum starved amoebic trophozoites harbouring GFP-EhRho5 were observed with a plain apochromatic 63x, 1.4NA oil immersion objective and a 488 nm laser. The trophozoites were stimulated with LPA for 15 mins (15μM, L7260), followed by wortmannin for 15 mins (100nM, W1628), according to the necessity of the experiment. Bleaching was performed during fly forward using ROI scan feature at 90–100% laser power. A spherical ROI was photobleached and subsequent images of the area was acquired every 1.6sec. Fluorescence of the ROIs over time was adjusted by background subtraction and photobleaching corrections. The recovery of fluorescence was determined by calculating the time required for 50% fluorescence recovery ($\tau_{1/2}$). The recovery curves were analysed with double exponential fit using the following equation in Cell Sens software (Olympus), leading to two recovery rates $\tau_1$ and $\tau_2$.

$$y = y_0 + A1.exp\left(-\frac{x}{\tau 1}\right) + A2.exp\left(-\frac{x}{\tau 2}\right)$$

Where A1 and A2 are the amplitudes during the fluorescence recovery, obtained by the double exponential fit. Similarly, τ1 and τ2 are the time constants obtained during the primary and secondary recovery of fluorescence in a double exponential model.

### Effector pulldown assay

The GTP-bound, "active" fractions of EhRho5 were determined essentially as described for Rac1 [109].

For *in vitro* binding assay, EhRho5 protein was incubated with excess of GMPPNP and GDP separately in presence of 10mM EDTA for faster exchange. As previously described, purified 0.1 μmole of GST-RBD/ PBD bound to Sepharose beads were incubated with 0.20 μmole exchanged EhRho5 for 30 min at room temperature [19,51]. Unbound protein was washed form the beads followed by sample preparation for Western blotting.

For checking Rho activation status inside cells, serum starved cells were stimulated with 15μM LPA for 15 mins, followed by 100nM wortmannin for 15mins at 37˚C Cells were then lysed in lysis buffer containing 10 mM $MgCl_2$ and 5 mM EDTA. lysates were incubated with bacterial expressed and purified GST- RBD/PBD bound to sepharose beads to capture active Rho/Rac. This mixture was incubated for 2 hrs at 4˚C and pelleted by centrifuge at 100 g for 5 mins. Pellet was then washed with lysis buffer without $MgCl_2$. Samples were prepared and proteins were resolved by SDS-PAGE followed by Western blotting using anti-HA monoclonal antibody to detect active Rho-GTPase.

### GEF assay

EhGEF2 activity for EhRho5 was determined using the method described previously [110]. Briefly, EhRho5 was mixed with 300nM fluorescent 2'(3')-bis-O-(N-methylanthraniloyl)-GDP and free mant-GDP was removed using NAP-5 column (cytiva). Nucleotide exchange reaction was initiated by addition of 100 μM GDP and varied concentrations (1–5μM) of EhGEF2 at 25˚C in a thermostatted cuvette, and the decrease in fluorescence was measured ($\lambda_{ex}$ = 360nm and $\lambda_{em}$ = 440 nm; slits = 5/5nm). After equilibration, GEF or buffer (uncatalyzed trace) is added. Observed pseudo first order exchange rate constants ($k_{obs}$) were obtained by nonlinear least square fit of data at each concentration of EhGEF2 to an exponential equation.

$$I(t) = (I_0 - I_\infty)exp\,(-k_{obs}t)\, + I_\infty$$

Here, I(t) is the intensity at time t, $I_0$ is the initial intensity and $I_\infty$ is the final intensity. Further, the catalytic efficiency was obtained from slope of a linear least square fit of the $k_{obs}$ values across EhGEF2 concentration.

$$k_{obs} = (k_{cat}/k_m)[EhGEF2] + k_{intr}$$

Where $k_{intr}$ is the intrinsic nucleotide exchange rate of the EhGEF2.

### Statistical analyses

For evaluation of datasets corresponding dextran uptake, and Western blot quantification, ratio-paired Student's *t* test were performed. Statistical significance in colocalization studies, translocation studies and FRAP were determined using two-tailed Student's *t* test. Super Plots were generated in GraphPad Prism, where each experiment is depicted in a colour. All the analyses were done in GraphPad Prism version 8.0 and the corresponding *p* values are indicated in the figure legends.

### Materials

LPA (15μM, Cat no. L7260, Sigma Aldrich), PDGF (60ng/μl; Cat no. P4306, Sigma Aldrich), Wortmannin (100nM, Cat no. W1628, Sigma Aldrich), anti-His monoclonal antibody (1:7000, Cat. no. MA1-21315, Invitrogen), G418 (6μg/ml, Cat no.1720, Sigma Aldrich), anti-HA

monoclonal antibody (1:130 (IF), Cat no. sc-7392, Santa Cruz Biotechnology), anti-Cysteine synthase polyclonal antibody (1:1000 (IB), kind gift from Dr. Tomoyoshi Nozaki), anti-Hgl monoclonal antibody (1:50(IB;3F4-7F4) 1:130(IF; 3F4), kind gift from Dr. William Petri), anti-HA monoclonal antibody (1:1000 (IB), C29F4, Cell signaling technology), anti-GFP polyclonal antibody (1:500 (IB), Roche) Texas Red Dextran (100μg/ml, Cat. D-1864, Life Technologies), Alexa568 labelled Phalloidin (1:40; Cat No. A12380; Thermo Fisher Scientific), CellTracker Orange CMRA Dye (2 μM; C34551), Mowiol (Cat. 81381-250G, Sigma Aldrich), Alexa Fluor 680 anti-rabbit antibody (1:10,000 (IB); A-21076; Thermo Fisher Scientific), Alexa Fluor 800 anti-mouse antibody (1:10,000 (IB); A- 32730; Thermo Fisher Scientific)

## Accession numbers

EHI_013260; EhRho1, EHI_012240; EhRho5, EHI_182740; EhGEF2, XP_656365.1; EhFP4.

## Supporting information

**S1 Fig.** (A) 200μg cell lysates from plasmid EhEx HA (control) and HA-EhRho5 trophozoites were resolved on SDS-PAGE and subjected to immunoblotting using anti-HA and anti-cysteine synthase antibodies. Cysteine synthase was used as loading control. (B) Immunofluorescence image of HA-EhRho5 trophozoites, using anti-HA antibody (Scale bar = 10μm). (C) Relative pixel intensities are plotted as SuperPlots for HA-EhRho5 trophozoites in presence of LPA at plasma membrane and endomembrane. Normalisation was performed with EhRho5 cytosolic intensity. Statistical significance was determined using unpaired Student's t-test (N = 3, n>80 (Plasma membrane), n≥40 (Endomembrane); $^{**}$p<0.01). (D) A new cell line expressing HA-EhRho5 trophozoites was examined for LPA stimulation associated changes. Dot plot shows the percentage of cell population exhibiting membrane associated EhRho5 in presence as well as in absence of LPA, in HA-EhRho5 trophozoites. Values are represented as mean ± SEM of three independent experiments (N = 3, n≥80; unpaired Student's t-test, $^{**}$p<0.01). (E) Quantitative analysis of relative pixel intensities in LPA treated cells compared to untreated in new cell line expressing HA-EhRho5. SuperPlot shows comparison of relative pixel intensities of HA-EhRho5 fluorescence at plasma membrane in LPA treated and untreated cells (N = 3, n≥80; unpaired Student's t-test, $^{*}$p<0.05). (F) 3D reconstruction of HA-EhRho5 trophozoites, post stimulation with and without LPA. Arrowheads show the membrane regions. (G) HA-EhRho5 serum starved trophozoites were stimulated with 60ng/μl PDGF. Cells were fixed and immunostained using anti-HA antibodies (Scale bar = 10μm). Quantification of relative pixel intensity at plasma membrane in presence and in absence of PDGF. Values were plotted as mean ± SEM in SuperPlots. N = 3, n = 85, unpaired Student's t-test, p>0.05. (H) Expression analysis of EhRho5 mRNA in presence and absence of LPA stimulation. Wild type trophozoites were stimulated with or without LPA and mRNA isolation followed by cDNA synthesis was performed. Semi-Q PCR was performed to assess the expression. (I) Expression analysis of EhRho5 protein in presence and absence of LPA stimulation. HA-EhRho5 trophozoites were stimulated with or without LPA, lysed and resolved on 12% SDS-PAGE. Immunoblotting was performed using anti-HA and anti-CS antibodies. (J) Additional immunoblot of surface biotinylation to capture membrane bound EhRho5. (K) 200μg cell lysates from plasmid EhEx HA (control) and HA-EhRho5CA trophozoites were resolved on SDS-PAGE and subjected to immunoblotting using anti-HA and anti-CS antibodies. Cysteine synthase was used as loading control.
(TIF)

**S2 Fig.** (A-C) Panel shows protein purification profiles of His-EhRho5, GST-RBD and GST-PBD respectively. Purified proteins are indicated with an arrow. (D) Additional immuno-blot of In vitro binding assay (ref Fig 2A). (E) Additional immunoblot of Rho activation assay (ref Fig 2C). (F) GFP-EhRho5 expressing trophozoites induced with 30μg/ml tetracycline for 48 hrs were lysed in lysis buffer. 300μg cell lysates were resolved on SDS-PAGE and subjected to immunoblotting using anti-GFP and anti-CS antibodies. Cysteine synthase and uninduced lysate were used as control. (G) A new cell line expressing GFP-EhRho5 trophozoites was examined for LPA stimulation associated changes. Serum starved GFP-EhRho5 trophozoites were stimulated with LPA. Representative image shows the localisation of GFP-EhRho5 in presence as well as in absence of LPA (Scale bar = 10μm). (H) Quantitative analysis of LPA treated cells compared to untreated. SuperPlot shows comparison of relative pixel intensities of GFP-EhRho5 fluorescence at plasma membrane and endomembrane (N = 3, n≥150; unpaired Student's t-test, ** $p < 0.01$). (I) Dot plot shows the percentage of cell population exhibiting membrane associated EhRho5 in presence as well as in absence of LPA, in GFP-EhRho5 trophozoites. Values are represented as mean ± SEM of three independent experiments (N = 3, n≥120; unpaired Student's t-test, ** $p < 0.01$). (J) Representation of fluorescence recovery time τ2 for indicated conditions.
(TIF)

**S3 Fig.** (A) Confirmation of Knockdown of EhRho5. The knockdown efficiency was deter-mined by Semi-Q PCR. Normalisation of EhRho5 expression was done with actin band inten-sity (N = 4, mean ± SEM, ratio paired Student's t-test, * $p < 0.05$). EhRacG (EhRho2; 55% identity with EhRho5) was used as control to check the specificity of knockdown. (B) RNAi mediated knockdown of EhRho5 affects growth of trophozoites. Trophozoites were grown for 5 days and every 24hrs cells were harvested and counted. (C, D) Roundness of the cells was cal-culated using ICY software and plotted. (N = 3, n>140, unpaired Student's t-test, ns (non-sig-nificant) $p > 0.05$, **** $< 0.0001$).
(TIF)

**S4 Fig.** (A) Relative pixel intensities are plotted as SuperPlots for LPA pre-treated GFP-EhRho5 trophozoites upon wortmannin or DMSO treatment. N = 3, n>140, mean ± SEM, unpaired Student's t-test, ** $p < 0.01$, * $p < 0.05$) (B) Panel shows additional immu-noblot (ref. to Fig 4C). (C) Serum starved LPA pre-treated HA-EhRho5 trophozoites were studied for their dextran uptake for indicated conditions. Trophozoites were treated with TR-Dextran along with wortmannin and DMSO, individually. Cells were fixed with 4% PFA and immunostained with anti-HA antibody, followed by image acquisition in a confocal microscope. CTCF was determined for each condition as described in Materials and Methods. Values represent means ± SEM, N = 4, n>60, ratio-paired Student's t-test * $p < 0.05$). (D) Representation of fluorescence recovery time τ2 for indicated conditions.
(TIF)

**S5 Fig.** (A-B) Purified His-tagged EhFP4 and EhGEF2, respectively, were resolved with SDS-PAGE followed by staining with Coomassie Blue. Arrows indicate the affinity purified pro-teins. (C) In vitro functional analysis of EhRho5. Nucleotide exchange kinetics of EhRho5 (2μM) in Mg2+ and EDTA buffer. (D) Exchange kinetics of EhRho1 (2μM) and EhRho5 (2μM) in pres-ence and in absence of EhGEF2 (1μM). A single replicate data trace is shown. (E) GEF activity exhibited by different concentrations of EhGEF2 for 2μM of EhRacG (EhRho2). Panel shows a single data trace. (F) Catalytic efficiency (kcat/km) was obtained from the slope of a linear least square fit of kobs values against EhGEF2 concentration. Values represent mean ± SEM of three independent measurements (N = 3). (G) Knockdown confirmation of EhGEF2. The knockdown efficiency was determined by Semi-Q PCR. EhGEF2 expression was normalised to actin (N = 3,

ratio paired Student's t-test, ***p<0.001). Specificity of knockdown was checked with expression of EhGEF1 (25% identity with EhGEF2) (H) 200μg cell lysate, each from EhEx HA (control) and HA-EhGEF2 trophozoites were resolved on SDS-PAGE and subjected to immunoblotting using anti-HA and anti-CS antibodies. (I) Representative images of HA-EhGEF2 overexpressing trophozoites compared to EhEx HA (control), scale bar = 10μm. (J) Representative 3D reconstructions of HA-EhGEF2 trophozoites, in presence or in absence of LPA (15μM). Arrowheads represent the macropinocytic cups. (K) Additional lipid blots show the binding of His-EhGEF2 to different phosphoinositides. (L) Confocal image exhibiting co-existence of HA-EhGEF2 and F-actin on macropinocytic cups, indicated by arrowheads (Scale bar = 10μm). (M) Representative confocal image of HA-EhGEF2 trophozoites performing TR-Dextran uptake. EhGEF2 is localised at the tips of macropinocytic cups, shown by arrowheads (Scale bar = 10μm).
(TIF)

**S1 Table. Parameters obtained from FRAP studies upon LPA stimulation. (Ref. Fig 2).**
(TIF)

**S2 Table. Parameters obtained from FRAP studies upon inhibitor treatment. (Ref. Fig 4).**
(TIF)

**S3 Table. Accession numbers of Rho GTPases used in the study.**
(TIF)

**S1 Video. Animation of 3D reconstruction for untreated (SS) HA-EhRho5 trophozoites and LPA stimulated (LPA) HA-EhRho5 trophozoites.**
(MOV)

**S2 Video. FRAP video showing recovery of fluorescence at plasma membrane in presence or in absence of LPA stimulation.**
(AVI)

**S3 Video. FRAP video showing recovery of fluorescence at endomembrane in presence or in absence of LPA stimulation.**
(AVI)

**S4 Video. Macropinocytosis in GFP-EhRho5 trophozoites using TR-Dextran.** Trophozoites were incubated with 2mg/ml TR-Dextran and immediately imaged using an Olympus confocal microscope, approximately 1.6sec/frame with 60x/1.4NA oil immersion objective.
(AVI)

**S5 Video. Random migration of GFP vector and GFP-EhRho5 overexpressing trophozoites.** Cells were incubated on a glass bottom dish for 30–45 mins. Videos were acquired using a live cell Olympus confocal microscope, for roughly 5 mins, with a time interval of 1.6sec between each frame. The display of the tracks uses the Temporal Rendering of the Track Manager which sets colors depending on the elapsed time.
(MOV)

**S6 Video. Random migration of EhRho5 depleted (4Trigger-EhRho5) and Wild-type trophozoites.** Cells were labelled with 2μM cell tracker orange for 30 mins followed by incubation on a glass bottom dish for 30–45 mins. Videos were acquired using a live cell Olympus confocal microscope, for roughly 5 mins, with a time interval of 1.6sec between each frame. The display of the tracks uses the Temporal Rendering of the Track Manager which sets colors depending on the elapsed time.
(MOV)

**S7 Video. FRAP video showing recovery of fluorescence at plasma membrane upon DMSO and wortmannin treatment in LPA pre-treated GFP-EhRho5 trophozoites.**
(AVI)

**S8 Video. FRAP video showing recovery of fluorescence at endomembrane upon DMSO and wortmannin treatment in LPA pre-treated GFP-EhRho5 trophozoites.**
(AVI)

**S9 Video. Animation of 3D reconstruction for untreated (SS) HA-EhGEF2 and LPA stimulated (LPA) HA-EhGEF2 trophozoites.**
(MOV)

# Acknowledgments

We are thankful to Dr. Tomoyoshi Nozaki (Graduate School of Medicine, University of Tokyo), Dr. Kumiko Nakada-Tsukui (Department of Parasitology, National Institute of Infectious Diseases, Tokyo Japan), Dr. Kuldeep Verma (Laboratory of Infectious Diseases, CDFD, India), Dr. Anil Raj Narooka (former graduate student) for their insightful suggestions regarding the project, and Ms. Megha Jain (former graduate student) for teaching FRAP assay. We also thank Dr. Saptarshi Mukherjee (Ultrafast and molecular spectroscopy laboratory, IISER Bhopal) and Dr. Ishu Saraogi (Department of Chemistry, IISER Bhopal) for the fluorometer instrumentation facility. We acknowledge the Fund for Improvement of Science and Technology Infrastructure (FIST) facility at IISER Bhopal by the Department of Science and Technology (DST) for live-cell microscopy and central instrumentation facility for confocal microscopy.

# Author Contributions

**Conceptualization:** Achala Apte, Elisabeth Labruyère, Sunando Datta.

**Data curation:** Achala Apte.

**Formal analysis:** Achala Apte, Maria Manich, Elisabeth Labruyère.

**Investigation:** Achala Apte, Sunando Datta.

**Methodology:** Achala Apte.

**Resources:** Sunando Datta.

**Supervision:** Sunando Datta.

**Writing – original draft:** Achala Apte.

**Writing – review & editing:** Achala Apte, Elisabeth Labruyère, Sunando Datta.

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
