## [Decision Letter · Decision Letter 0]

29 Jan 2022

Dear Dr. Datta,

Thank you very much for submitting your manuscript "PI3K-GEF2-EhRho5 axis contributes to LPA stimulated macropinocytosis in E. histolytica" for consideration at PLOS Pathogens. As with all papers reviewed by the journal, your manuscript was reviewed by members of the editorial board and by several independent reviewers. In light of the reviews (below this email), we would like to invite the resubmission of a significantly-revised version that takes into account the reviewers' comments.

The major criticisms by the second referee should be adequately addressed.

We cannot make any decision about publication until we have seen the revised manuscript and your response to the reviewers' comments. Your revised manuscript is also likely to be sent to reviewers for further evaluation.

Sincerely,

Dominique Soldati-Favre

Section Editor

PLOS Pathogens

Dominique Soldati-Favre

Section Editor

PLOS Pathogens

Kasturi Haldar

Editor-in-Chief

PLOS Pathogens

orcid.org/0000-0001-5065-158X

Michael Malim

Editor-in-Chief

PLOS Pathogens

orcid.org/0000-0002-7699-2064

The major criticisms by the second referee should be adequately addressed.

Reviewer's Responses to Questions

**Part I - Summary**

Reviewer #1: Apte et al present a study that demonstrates the roles of a Rho5-like protein (EhRho5), its GEF, and PI3 kinase in endocytosis and motility in in the protozoan parasite, Entamoeba histolytica. They show that EhRho5 and its GEF translocate to the plasma membrane, macropinosomes, and other internal membranes upon stimulation, that genetic reduction of EhRho5 or its GEF reduces macropinocytosis and increases motility, and that treatment with a PI3K inhibitor phenocopies the mutants. Endocytosis and motility are important to virulence in E. histolytica. Therefore, the work is significant because it provides insight into pathogenicity. The work exhibits several strengths including its comprehensiveness, its use of state-of-the-art methods, and its extensive use of controls in experiments.

Reviewer #2: In this manuscript, the authors described constitutive and Lysophosphatidic acid (LPA) stimulated macropinocytosis in Entamoeba histolytica trophozoites functions via the PI3K-EhGEF2-EhRho5 axis. Macropinocytosis is one of the major processes responsible for amoebic sustenance in the host and pathogenicity.

It has been well established that Rho GTPases modulate the actin cytoskeletal dynamic and Rho activation by Rho-specific guanine nucleotide exchange factor (GEF). So far, EhRho5 (EHI_012240, EhRacD) has been shown to be associated with phagosomes, and EhGEF2 has been reported as potential GEF candidates for EhRho5. Here, the authors have demonstrated the involvement of EhRho5 in constitutive and LPA stimulated macropinocytosis. They also report that LPA, a bioactive phospholipid, activates EhRho5 and translocates it from the cytosol to the plasma membrane and endomembrane compartments. Moreover, they demonstrated that PI3K is involved in LPA stimulated translocation and activation of EhRho5, leading to macropinocytosis. They further identified EhGEF2 as a guanine nucleotide exchange factor of EhRho5.

This study was conducted carefully planned and executed. However, there are some significant concerns stated below in "Major Points".

**Part II – Major Issues: Key Experiments Required for Acceptance**

Reviewer #1: (No Response)

Reviewer #2: The main question in this study is unclear. The molecular mechanism of signaling to stimulate macropinocytosis is an important and interesting topic to study, however, the importance of studying LPA stimulated macropinocytosis in this parasite is not fully explained. Furthermore, the authors stated macropinocytosis in this parasite is stimulated via the PI3K-GEF2-Rho5 axis without explaining the uniqueness and conservation of this signal in model organisms. What is the significance of LPA and its signaling in this parasite? How does this link to pathogenesis? Please clarify the reason why the authors believe LPA and PDGF mediated Rho activation signaling pathway exists in E. histolytica. What is the physiological state the authors hypothesized? Related to these points, the difference between LPA and PDGF did not explain. Please reconsider the story to discuss and clarify these points

The nomenclature of each Rho GTPase is so confusing. Please follow the previous name: EHI_012240 as EhRacD and EhRho1 as EHI_029020, not EHI_013260.

From line 321, it is shown that EhGEF2 activates EhRacG as a preferential substrate among the Rho family GTPases. To emphasize the importance of EhGEF2 for EhRho5, it is preferable to show how much the activation of EhRho5 by EhGEF2 occurs in vivo or in the presence of EhRacG.

As stated in the introduction, EhRho5 is found in the phagosome proteome. How do authors find the main role of EhRho5 is in macropinocytosis?

What is the definition of micropinocytosis and pinocytosis? The authors used dextran uptake as a macropinocytosis model. However, the efficiency of the dextran uptake is explained as “pinocytic efficiency”. Pinocytosis is generally used for selective uptake mediated by clathrin and/or caveolae and the size of internalized vesicle is small (<150 nm). In contrast, macropinocytosis is non-selective and receptor independent process generate larger vesicle (0.2~5 μm). It has better to use “macropinocytic efficiency”. Related to this, images to show macropinosomes in figures 3 and 5 are poor. Most of the trophozoites did not acquire dextran and sometimes arrows indicated extracellular aggregate. Please show clear images.

Authors showed EhRho5 preferentially bound to RBD, not to PBD. Please discuss this point based on the amino acid sequence of EhRho5.

Localization of HA-EhRho5 and GFP-EhRho5 is significantly different. In the steady-state, HA-EhRho5 and GFP-EhRho5 are preferentially localized in the cytoplasm and on the membrane, respectively. The authors used HA-EhRho5 to show LPA stimulated translocation to membrane and GFP-EhRho5 to show the opposite effect by Wortmannin. It is not fair. It is necessary to examine the localization of endogenous EhRho5 by using a specific antibody detecting endogenous protein.

It is unclear what molecule is responsible for tethering EhRho5 on the plasma membrane shown by biotinylation. Please include this point in the discussion.

Specificity of expression gene silencing is unclear. In EhRho5 and EhGEF2 silenced strain, please evaluate the level of another Rho and RhoGEF especially share a high identity. Adding to this, please explain more about the method for silencing. pKT3M vector is the name of the vector to express a myc-tagged protein in E. histolytica. Change the name of this vector appropriately. Include the information of what trigger sequence is in the plasmid. Does “WT control” means mock vector transformant or untransfected trophozoites?

Please add more explanation for the FRAP assay. It is unclear what tau1 and tau2 mean. Also, please discuss whether this phenotype is general characteristics of Rho or EhRho5 specific.

Authors claimed that EhGEF2 is the EhoGEF for EhRho5. This hypothesis comes from a precedence study. There are over 60 of RhoGEFs in E. histolytica. Please add discussion to explain the adequacy of this hypothesis. Also, to establish the PI3K-EhGEF2-EhRho5 axis, EhGEF2 should recognize 3-phosphorylated phosphatidylinositol, as discussed in the discussion and shown in Figure 6. In ref #54, the authors of the paper showed that recombinant EhGEF2 bound to phosphatidic acid but not the other phosphoinositides examined, such as PI(3)P, PIP2, and PIP3. Please explain the hypothesis include this result.

**Part III – Minor Issues: Editorial and Data Presentation Modifications**

Reviewer #1: 1. Line 88: The authors mention that the overexpressor is made using an “endogenous amoebic promoter.” This is misleading. Endogenous implies that there was an integration event into a chromosome so that expression is driven from a natural promoter. But, as far as this reviewer can tell, the overexpression plasmids remain episomal. The authors should clarify this point.

2. Line 105: To confirm plasma membrane localization of EhRho5 after stimulation, the authors stimulate cells, biotinylate the cell surface, and then purify labeled components with avidin. The authors show that EhRho5 is enriched in this purified fraction. In general, Rho proteins would not appear on the exoplasmic face of cells. Thus, they would not normally be exposed to be biotinylated. Therefore, how do the authors think the enrichment is happening? Is it because EhRho5 is interacting with another protein that would be exposed on the cell surface? The authors should discuss this point.

3. It is not clear what is gained by showing the microscopy images in Figure 3. For example, there isn’t any difference between -LPA and +LPA in 3A. There are minimal differences between cell lines in the TR-Dextran panels in 3C and 3E. Even though the panel for TR-Dextran for the HA-EhRho5-CA shows two endosomes (vs one endosome for the other cell lines), the authors are only showing one cell and not a field of cells. The quantification (3B,D,F,H) and the accompanying videos are sufficient support for the conclusions.

4. The technical abstract does would not stand alone in a database such as PubMed. There is no mention of the pathogen until the last line. The whole genus name is not mentioned. There is no information about the importance of this organism. Readers outside the field, who arrive at this abstract in a database, would not fully understand the importance of the study.

5. The authors mention in Line 43 of the Introduction that E. histolytica can take up 15% of cell volume. In what time frame (1 hour? 1 day?)? The information doesn’t mean much unless a time frame is given.

6. In several instances in the Introduction and Discussion, the authors call Dictyostelium a “homologue” of E. histolytica. Organisms cannot be homologues of one another. They can be genetically related.

7. Line 187: The authors state the “EhRho5 trophozoites showed no significant change in pinocytic efficiency…” Do the authors mean HA-EhRho5 trophozoites?

8. Throughout the Methods the authors use % as a unit (e.g., heat inactive serum, paraformaldehyde, triton-x-100, fbs, skim milk etc.). The reviewer assumes this is % (v/v), but the authors should include the complete unit.

9. Line 430: The authors mention using tetracycline to induce expression. It’s not clear which of the Entamoeba expression vectors, used in this study, are tet-inducible and which ones are not tet-inducible. When did the authors use tetracycline to induce expression and what proteins? There is no mention of tetracycline induction when describing the experiments in the Results. This information is important because it helps to reveal if the correct controls are used (e.g., irrelevant tet-inducible protein or empty plasmid).

10. In the Methods, the Western blotting protocol is not adequately described. For example, how long did the authors incubate with primary and secondary antibody? How many washes? How were the blots developed?

11. The Methods are lacking the information on how cells were treated with LPA, PDGF, or wortmannin. Although the information is in the figure legends it should be included in the Methods.

12. In the Methods and Acknowledgements, the authors thank a number of individuals for reagents and helpful suggestions. However, the authors never present the affiliations of these other scientists. The authors should include this information.

Reviewer #2: In line 28, “dysentery, abscesses and diarrhoea” would be “dysentery, diarrhoea, and abscess formation”.

In line 42, refer some review article would be appropriate.

In line 61 and others, Dictyostellium should be Dictyostelium and also it is not be a “homologue” of E. histolytica. Please use appropriate explanation.

In line 68 and others, please discuss the fact in E. histolytica and other model organisms in a clearly separated way. The references cited here are a mixture of the observations from this amoeba and other organisms. It is also applicable in the following text.

In line 88, please clarify what promoter used to express EhRho5.

In line 93, to quantify the membranous localization of HA-EhRho5 in Fig. 1B, how to define the “membrane”? Please clarify.

In line 97, “Gal-Gal Lectin” should be “Gal/GalNAc specific lection”. Please confirm.

In fig.1C, cross section is applicable only a part of the cell. Please show the section of entire cell.

In line108, Neutravidin is a name of the product with trademark. It should be written as NeutrAvidinTM.

In line 130, please add an explanation for adequacy to use representative Rho and Rac binding domains to E. histolytica Rho.

In line 148, it is indicated as ‘Fig. 2B’, but the effect of LPA stimulation is indicated in ‘Fig. 2D’.

In Fig. 3C, please depict the cell morphology at the middle panel for EhExHA.

In Fig. 3C-H, it is unclear whether authors evaluated the macropinocytosis efficiency in HA- and GFP- expressed amoebas or all amoebas in the field. Please clarify.

In line 222, it has better to refer original article to show PI3K inhibition by wortmannin.

In line 258, please spell out “PIP3”.

In line 293, it has better to cite original article related to this topic.

In line 400, “BIS33 medium” or “BI-S-33 media”? Please be consistent.

In line 430, which transfectants are controlled by a tetracycline-inducible system is not clear. Please clarify.

In line 439, please describe references of antibodies used in this study.

In line 440, secondary antibodies for Western blotting are conjugated with which Alexa Fluor dye? Please clarify. Also, please add detection method of the image.

In line 446, anti-HGL should be anti-Hgl. Also, in the Western blotting, there are two kinds for this antibody were used. Which one has used for this assay?

In line 449, please add provider for the Mowiol.

In line 463, what is the BIS33 incomplete media? “S” in the BI-S-33 medium stands for “serum”. If the “incomplete” mean serum-free medium, it should explain as BI medium. Or clarify what is missing in the incomplete medium.

In line 471~, please clarify the logic and calculation of how to measure the migration speed.

In line 482~, plasmid construction method for pGEX plasmids was missing.

In line 491, the detail for affinity purification is missing. It should be clarified to understand the recombinant protein purification process shown in Fig. S2A~C and Fig. S5A and B.

In line 514 and 849, the recovery time is indicated as τ1/2, but in other parts, such as line 160 and line 162, the recovery time is indicated as t1/2. Please confirm.

In line 502m, “western blotting” should be “Western blotting”. The same applied to the entire text.

In line 506, “GFP-Rho5” should be “GFP-EhRho5”.

In line 525, 0.1μMole should be 0.1 μmole. Also, add references for GST-RBD/PBD constructs.

In line 563, “anti-mouse His” should be “anti-His monoclonal antibody”.

In line 564, “HA antibody” should be an “anti-HA monoclonal antibody”. The same applied to the expression of all the other antibodies, too.

In Fig.6, the LPA receptor is not discussed in the main text. Please include. It has been shown that PI3P is on the endomembrane not on the plasma membrane in E. histolytica.

In figure legends, please state what showed in each panel, then explain the brief experimental procedure.

In Fig. S5E, the image of anti-actin stating should be in the lower panel, same as in Fig. S3A.

PLOS authors have the option to publish the peer review history of their article (what does this mean?). If published, this will include your full peer review and any attached files.

Reviewer #1: No

Reviewer #2: No
---

## [Editor Report · Decision Letter 1]

26 Apr 2022

Dear Dr. Datta,

We are pleased to inform you that your manuscript 'PI Kinase-EhGEF2-EhRho5 axis contributes to LPA stimulated macropinocytosis in Entamoeba histolytica' has been provisionally accepted for publication in PLOS Pathogens.

Best regards,

Tomoyoshi Nozaki, M.D., Ph.D.

Associate Editor

PLOS Pathogens

Dominique Soldati-Favre

Section Editor

PLOS Pathogens

Kasturi Haldar

Editor-in-Chief

PLOS Pathogens

orcid.org/0000-0001-5065-158X

Michael Malim

Editor-in-Chief

PLOS Pathogens

orcid.org/0000-0002-7699-2064
---

## [Editor Report · Acceptance letter]

18 May 2022

Dear Dr. Datta,

We are delighted to inform you that your manuscript, "PI Kinase-EhGEF2-EhRho5 axis contributes to LPA stimulated macropinocytosis in Entamoeba histolytica," has been formally accepted for publication in PLOS Pathogens.

Best regards,

Kasturi Haldar

Editor-in-Chief

PLOS Pathogens

orcid.org/0000-0001-5065-158X

Michael Malim

Editor-in-Chief

PLOS Pathogens

orcid.org/0000-0002-7699-2064